# Lowering the Cu-O bond energy in CuO nanocatalysts enhances the efficiency of NH$_3$ oxidation

Lu Chen [1,2,3] ✉, Xuze Guan[2], Zhangyi Yao[2], Shusaku Hayama[4], Matthijs A. van Spronsen [4], Burcu Karagoz [4], Georg Held [4], David G. Hopkinson [5], Christopher S. Allen [5,6], June Callison [7], Paul J. Dyson [8] ✉ & Feng Ryan Wang [2] ✉

Tuning the electronic properties of nanocatalysts via doping with mono-dispersed hetero-metal atoms is an effective method used to enhance catalytic properties. Doping CuO nanoparticles with monodispersed Co atoms using different reductants affords catalysts (Co$_B$Cu/Al$_2$O$_3$ and Co$_H$Cu/Al$_2$O$_3$) with strikingly different electronic structures. Compared to Co$_H$Cu/Al$_2$O$_3$, the CuO nanoparticles in Co$_B$Cu/Al$_2$O$_3$ have longer and weaker Cu-O bonds, with a lower $1s \rightarrow 4p_z$ antibonding transition and higher $4p \rightarrow 1s$ bonding transition (as demonstrated from HERFD-XANES and valence-to-core X-ray emission spectroscopy). The weaker Cu-O bonds in Co$_B$Cu/Al$_2$O$_3$ lead to superior redox activity of the CuO nanoparticles, evidenced from *operando* XAFS and in-situ near ambient pressure-near edge X-ray absorption fine structures studies. Such superior redox properties of CuO in Co$_B$Cu/Al$_2$O$_3$ result in a much reduced activation energy of Co$_B$Cu/Al$_2$O$_3$ compared to Co$_H$Cu/Al$_2$O$_3$ (40.0 vs. 63.5 kJ/mol), thus leading to an enhancement in catalytic performance in the selective catalytic oxidation of NH$_3$ to N$_2$.

Ammonia (NH$_3$) emissions are projected to increase in the future, driven by its expanding use as a sustainable fuel, particularly in the shipping industry, as well as emissions from industrial processes[1,2]. Since NH$_3$ emissions are far less regulated than fossil fuel-based emissions, they are an important driver for fine particulate matter (PM 2.5) pollution[3,4]. Hence, the selective catalytic oxidation (SCO) of NH$_3$ to N$_2$ is a critical and increasingly used technology for mitigating NH$_3$ emissions that are detrimental to atmospheric quality[5]. This is because NH$_3$-assisted selective catalytic reduction (SCR) is the most widely used technology to reduce nitrogen oxide (NO$_x$) emissions from coal-fired power stations and diesel engines[6–8]. Achieving complete NO$_x$ removal necessitates an excess of NH$_3$, which results in NH$_3$ emissions (also referred to as NH$_3$ slip). The completion of an NH$_3$-SCO cycle comprises several elementary reaction steps that depend on both the properties of the catalysts and the reaction conditions, and are crucial in controlling the selectivity to N$_2$. Three established mechanisms for different catalysts in the NH$_3$-SCO reaction include the internal SCR (i-SCR) mechanism, the imide mechanism, and the hydrazine mechanism[9–11]. Among these, the i-SCR mechanism is predominantly applied to elucidate the NH$_3$-SCO reaction pathway of Cu-based catalysts[12–16], known for their cost-effectiveness and high selectivity.

[1]Low-Carbon Conversion Science and Engineering Center, Shanghai Advanced Research Institute, Chinese Academy of Sciences, Shanghai, China. [2]Department of Chemical Engineering, University College London, Roberts Building, Torrington Place, London, UK. [3]Yusuf Hamied Department of Chemistry, University of Cambridge, Cambridge, UK. [4]Diamond Light Source Ltd., Harwell Science and Innovation Campus, Chilton, Didcot, UK. [5]electron Physical Science Imaging Centre, Diamond Light Source Ltd., Didcot, UK. [6]Department of Materials, University of Oxford, Oxford, UK. [7]UK Catalysis Hub, Research Complex at Harwell (RCaH), Rutherford Appleton Laboratory, Harwell, UK. [8]Institute of Chemical Sciences and Engineering, École Polytechnique Fedérale de Lausanne (EPFL), Lausanne, Switzerland. ✉e-mail: chenlu@sari.ac.cn; paul.dyson@epfl.ch; ryan.wang@ucl.ac.uk

In the i-SCR mechanism, $NH_3$ undergoes initial oxidation to form NO (Step 1), and subsequently, the formed NO reacts with $NH_3$, yielding $N_2$ (SCR, Step 2). Step 1 involves redox processes of the metal catalyst and comprises the rate-determining step in $NH_3$ oxidation[16]. Supported CuO nanoparticles (NPs) exhibit high activity and selectivity for the subsequent SCR step[17–21], but are less active for Step 1. In contrast, supported noble metals and $Co_3O_4$ NPs exhibit high activity in Step 1, but are less efficient in the SCR step (Step 2)[22–26]. This imbalance can be overcome by integrating two metals to form a bimetallic catalyst to coordinate the sequential reactions, resulting in enhanced activity and selectivity for the entire $NH_3$-SCO reaction. Several bimetallic NP catalysts have been explored for the $NH_3$-SCO reaction, including those based on PtCu[27–29], AgCu[30,31], AuCu[32], and RuCu.[33] Despite the high activity of noble metal-containing catalysts, their high cost and low selectivity limit practical applications. Earth-abundant $CoO_x$ offers a cost-effective alternative, with the introduction of Cu into $CoO_x$ improving the $N_2$ selectivity of $CoO_x$-based catalysts[13,34].

$$Step1 : 4NH_3 + 5O_2 \rightarrow 4NO + 6H_2O$$

$$Step2 : 4NO + 4NH_3 + O_2 \rightarrow 4N_2 + 6H_2O$$

$$4NO + 4NH_3 + 3O_2 \rightarrow 4N_2O + 6H_2O$$

The activity and the selectivity of bimetallic catalysts are significantly influenced by their geometric and electronic structures[35–37]. The different electronic structures lead to differences in the strength of the metal−oxygen bond, which determines the distribution of ammonia oxidation products[38,39]. Oxides with high metal−oxygen bond strengths exhibit lower rates of reaction and facilitate a high selectivity to $N_2$. In contrast, metal oxides with weak metal−oxygen bond strengths lead to the formation of $NO_x$ (NO and $N_2O$). The unique structural features of single-site doped bimetallic NPs allow their electronic properties to be tuned more precisely than their monometallic counterparts, which provides a facile approach to modify the metal−oxygen bond strength in order to optimize both activity and $N_2$ selectivity[40]. Note that it has previously been shown that the nature of the reducing agent impacts the catalyst structure and activity[41,42].

In this work, the electronic structure of two bimetallic catalysts, $Co_BCu/Al_2O_3$ and $Co_HCu/Al_2O_3$, prepared using $NaBH_4$ and $H_2$, respectively, is studied with a range of X-ray spectroscopic techniques. Despite with same chemical composition, $Co_BCu/Al_2O_3$ exhibits enhanced activity in the $NH_3$-SCO reaction across all temperatures, surpassing the $Co_HCu/Al_2O_3$ catalyst. *Operando* X-ray absorption fine structure (XAFS) studies, combined with in-situ near ambient pressure-near edge X-ray absorption fine structure (NAP-NEXAFS) studies and *operando* diffuse reflectance infrared Fourier transform spectroscopy (DRIFTS), provide a detailed understanding of the electronic structure of the CuO NPs modulated by the single Co sites and the resulting impact on catalysis.

## Results and Discussion
### Synthesis and structural characterization of the catalysts
The bimetallic $Co_BCu/Al_2O_3$ catalyst (5 wt% Cu, 0.1 wt% Co) was synthesized by reducing $Cu(NO_3)_2 \cdot 3H_2O$ and $Co(NO_3)_2 \cdot 6H_2O$ together with $NaBH_4$. $Co_HCu/Al_2O_3$, with the same chemical composition, was synthesized by wet impregnation, using $H_2$ as a reductant. The Cu NPs are oxidized to CuO NPs when exposed to air. As shown in bright-field (BF) images and high-angle annular dark-field (HAADF) images from scanning transmission electron microscopy (STEM), the average size of the CuO NPs in $Co_BCu/Al_2O_3$ is 2.6 nm, whereas in $Co_HCu/Al_2O_3$ the average size is 2.0 nm (Fig. 1a−d, Supplementary Figs. S1 and S2). The size of the NPs hardly changes after catalysis for both $Co_BCu/Al_2O_3$ (Fig. S3) and $Co_HCu/Al_2O_3$ (Fig. S4). The interplanar spacing of the CuO lattice in $Co_BCu/Al_2O_3$ (2.0 Å) is slightly longer than in $Co_HCu/Al_2O_3$ (1.9 Å), indicating potentially different Cu-O bond lengths in the two catalysts (Fig. 1a−d). An energy-dispersive spectrometry (EDS) map of $Co_BCu/Al_2O_3$ shows a uniform elemental distribution confirming the presence of Cu and Co (Fig. S5).

Electron paramagnetic resonance (EPR) spectroscopy also confirmed that the major Cu species in both catalysts comprise NPs and not monodispersed Cu sites (Fig. S6). X-ray diffraction patterns of $Co_BCu/Al_2O_3$ and $Co_HCu/Al_2O_3$ show no obvious distinctions from the $Al_2O_3$ support (PDF #10-0425) (Fig. S7a), indicative of small NPs with a uniform size distribution. $H_2$-temperature programmed reduction (TPR) confirms that incorporation of Co atoms in $Co_BCu/Al_2O_3$ shifts the reduction temperature of CuO NPs by 34 °C (first peak) to a lower temperature (Fig. S8), indicating that Co promotes the reduction of

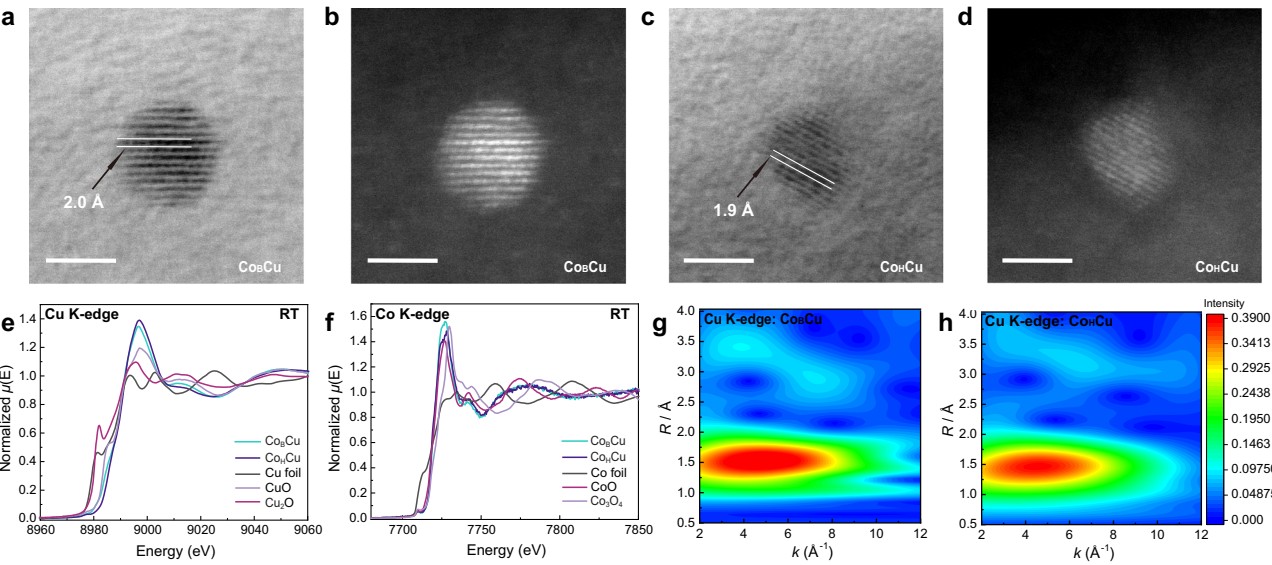

**Fig. 1 | Characterization of $Co_BCu/Al_2O_3$ and $Co_HCu/Al_2O_3$.** BF images (**a**) and HAADF image (**b**) of $Co_BCu/Al_2O_3$ (Scale bar = 2 nm); BF image (**c**) and HADDF image (**d**) of $Co_BCu/Al_2O_3$ (Scale bar = 2 nm); Cu K-edge HERFD-XANES (**e**) and Co K-edge XANES (**f**) of $Co_BCu/Al_2O_3$ and $Co_HCu/Al_2O_3$; 2D WT-EXAFS maps of the CuO NPs in $Co_BCu/Al_2O_3$ (**g**) and $Co_HCu/Al_2O_3$ (**h**).

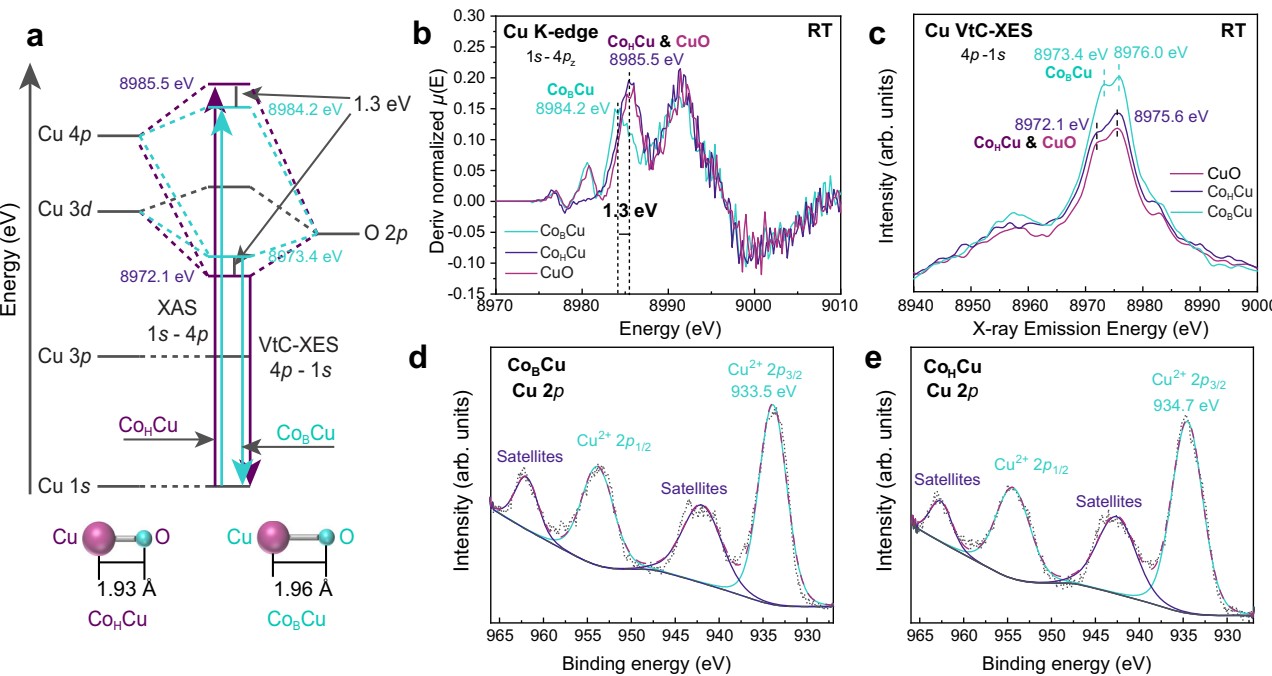

**Fig. 2 | Characterization of Co_BCu/Al_2O_3 and Co_HCu/Al_2O_3. a** simplified molecular orbital diagram of the CuO NPs in Co_BCu/Al_2O_3 and Co_HCu/Al_2O_3; **b** first derivative XANES spectra and **c** VtC-XES of Co_BCu/Al_2O_3 Co_HCu/Al_2O_3, and CuO/Al_2O_3; Cu 2p XPS spectra of Co_BCu/Al_2O_3 (**d**) and Co_HCu/Al_2O_3 (**e**).

CuO, probably via the asymmetric Co-O-Cu bond[43]. In contrast, for Co_HCu/Al_2O_3, two separate peaks indicate weaker interactions between the Cu and Co species. The broad peak in the T range of 200–300 °C of Co_BCu/Al_2O_3 in H_2-TPR may originate from bulk CoO_x, due to the aggregation of Co under H_2.

The chemical environments of the Cu species in Co_BCu/Al_2O_3 and Co_HCu/Al_2O_3 are different, as determined from extended X-ray absorption fine structure (EXAFS), X-ray absorption near-edge structure (XANES), and high-energy resolution fluorescence detected - X-ray absorption near-edge structure (HERFD-XANES) measurements (Fig. 1e, f). The major Cu species in both Co_BCu/Al_2O_3 and Co_HCu/Al_2O_3 are Cu^{2+}, as expected for CuO-based NPs, and the Co species in both Co_BCu/Al_2O_3 and Co_HCu/Al_2O_3 are also in the Co^{2+} oxidation state. The fitted EXAFS data reveal differences between the electronic structures of Co_BCu/Al_2O_3 and Co_HCu/Al_2O_3 (Table S1, Figs. S9–S11). Co_BCu/Al_2O_3 and Co_HCu/Al_2O_3 have a similar Cu-O coordination number (C.N.) of 3.45 ± 0.12 and 3.33 ± 0.12, respectively, and a similar Co-O C.N. of 5.74 ± 0.99 and 5.92 ± 1.09, respectively. However, Co_BCu/Al_2O_3 has a larger Cu-Cu (1) C.N. of 2.39 ± 0.48 and a Cu-Cu (2) C.N. of 1.90 ± 0.39, compared to Co_HCu/Al_2O_3 with a Cu-Cu (1) C.N. of 0.22 ± 0.21 and Cu-Cu (2) C.N. of 0. Moreover, the Cu-O bond length in Co_BCu/Al_2O_3 (1.96 Å ± 0.005) is longer than in Co_HCu/Al_2O_3 (1.93 Å ± 0.003), which is consistent with the TEM results. It is noteworthy that Co-Co bonds are not observed in either catalyst, indicative of single Co sites. Wavelet transform (WT) analysis of the EXAFS spectra leads to a 2D representation of the EXAFS (Fig. 1g, h), simultaneously revealing the signal features in both R- and k-space. The first shell peak with the most intense signal in Co_BCu/Al_2O_3 (centered at 4.7 Å^{-1}, 1.52 Å) is higher than the equivalent peak in Co_HCu/Al_2O_3 (centered at 4.5 Å^{-1}, 1.46 Å). This difference further confirms that Co_BCu/Al_2O_3 has longer Cu-O bonds, as the first shell is assigned to oxygen atoms coordinated to the copper.

The electronic structures of Co_BCu/Al_2O_3 and Co_HCu/Al_2O_3 were established using HERFD-XANES, valence-to-core X-ray emission spectroscopy (VtC-XES) and fine-scanned X-ray photoelectron spectroscopy (XPS) measurements (Figs. 2, S7). First derivative XANES contains a 1s → 4p_z transition at 8984.2 eV for Co_BCu/Al_2O_3 (Fig. 2b),

which is 1.3 eV lower than in Co_HCu/Al_2O_3 and CuO/Al_2O_3. In previous studies[16,44], the absorption energy for the 1s → 4p_z transition was shown to increase as the CuO loading decreases (from NPs to atomic sites), whereas a decrease in the 1s → 4p_z transition energy is rarely observed. This discrepancy suggests a different local coordination environment of Cu^{2+} in Co_BCu/Al_2O_3 (Fig. 2b). Furthermore, VtC-XES reveals that the CuO NPs in Co_BCu/Al_2O_3 have a 1.3 eV higher Cu K_{β2,5} feature (mainly 4p→1s transitions) than the CuO NPs in Co_HCu/Al_2O_3 and CuO/Al_2O_3 (Fig. 2c). The distribution of Cu 4p states is markedly influenced by the hybridization between Cu 3 d and ligand p states[45]. For Cu^{2+} species, the low-energy peak observed in the main K_{β2,5} line primarily arises from the contribution of the π-bonding state[45], which results from the hybridization of Cu 3d, O 2p and Cu 4p orbitals. The lower 1s→4p_z transition energy and higher π-bonding state of Co_BCu/Al_2O_3 point to different occupied and unoccupied states (Fig. 2a). Additionally, XPS of Co_BCu/Al_2O_3, Co_HCu/Al_2O_3, CuO/Al_2O_3, and CoCu/Al_2O_3 exhibit Cu^{2+} 2p_{3/2} peaks at 933.5, 934.7, 934.6, and 934.7 eV, respectively (Figs. 2d, e, S7c, d). Thus, the electronic states of Cu in Cu_HCo, CuO/Al_2O_3, and CoCu/Al_2O_3 are similar, with that of Cu_BCo being lower (Fig. S7c). The combination of HERFD-XANES, VtC-XES, and XPS suggests that the Cu-O bonds in the Co_BCu/Al_2O_3 catalyst are weakened relative to those observed in Co_HCu/Al_2O_3 (Fig. 2a), which is as expected, as the Cu-O bonds in Co_BCu/Al_2O_3 are longer (Fig. 1g, h). This feature indicates that the Cu-O bonds in Co_BCu/Al_2O_3 will be more reactive toward NH_3, leading to the reduction of Cu^{2+} and concomitant oxidation of NH_3. As proven in our recent publication, such a reaction is the rate-limiting step in the i-SCR mechanism[46]. Literature further supports this by showing that metal oxides with high metal–oxygen bond strengths exhibit lower rates in the NH_3-SCO reaction[47].

## Evaluation of the catalysts in the NH_3-SCO reaction
The catalytic performance of the bimetallic Co_BCu/Al_2O_3 and Co_HCu/Al_2O_3 catalysts was compared with single metal CuO/Al_2O_3 and CoO_x/Al_2O_3 catalysts in the NH_3-SCO reaction (Figs. 3a, S12 and S13). 5000 ppm NH_3 and GHSV 100,000 h^{-1} are used as the standard test conditions, which are consistent with waste gas streams in industrial processes and have been used in previous studies[48–51].

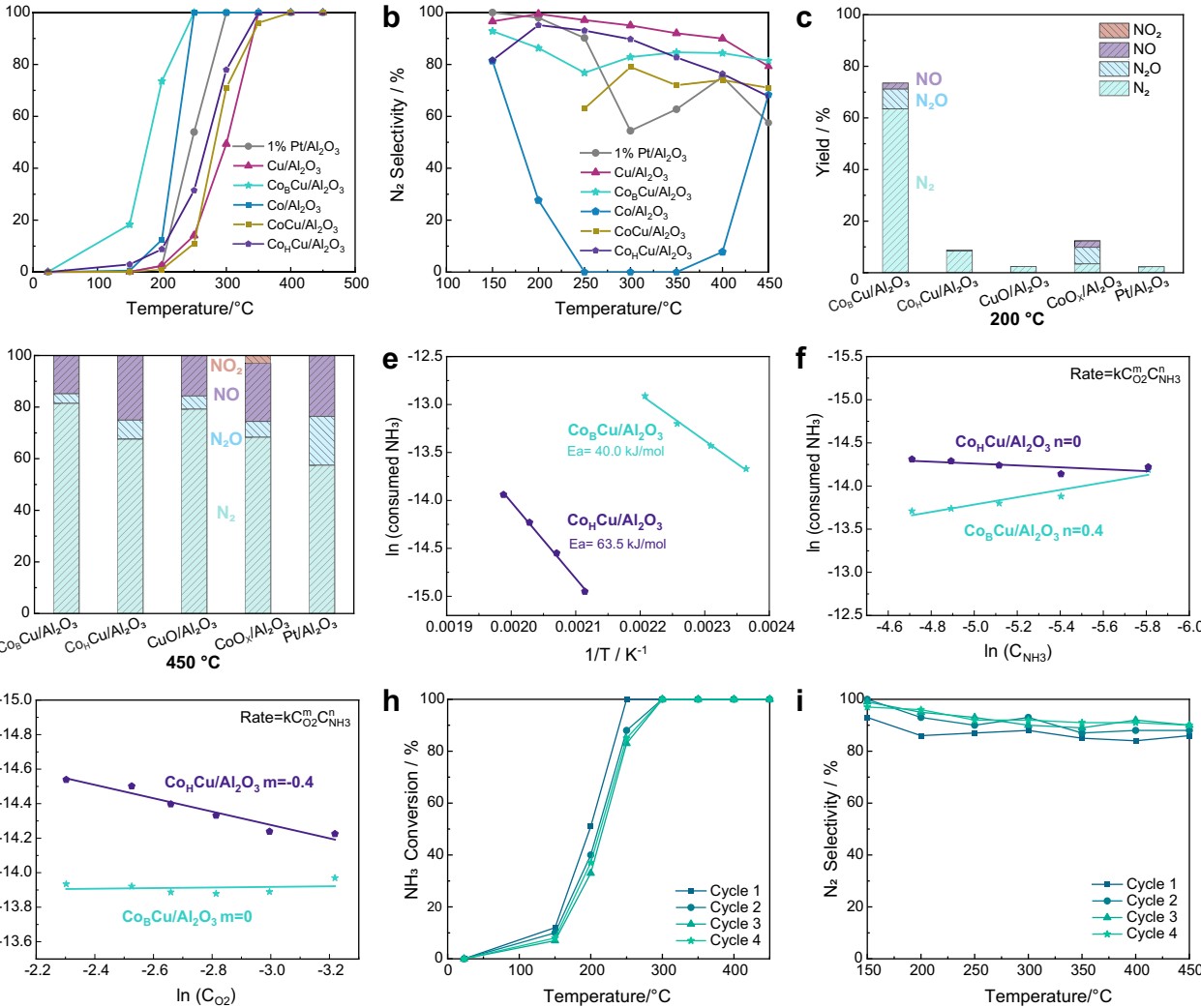

**Fig. 3 | Evaluation of $Co_BCu/Al_2O_3$, $Co_HCu/Al_2O_3$ and control catalysts in the $NH_3$-SCO reaction.** $NH_3$ conversion (**a**) and selectivity to $N_2$ (**b**) as a function of temperature (note that $CoCu/Al_2O_3$ corresponds to the catalyst without reduction); Product distribution at 200 °C (**c**) and 450 °C (**d**); **e** Activation energy for the $Co_BCu/Al_2O_3$ and $Co_HCu/Al_2O_3$ catalyzed reaction; Reaction order for $NH_3$ (**f**) and $O_2$ (**g**); Stability tests of $Co_BCu/Al_2O_3$ over 4 cycles (**h, i**) (50 mg $Co_BCu/Al_2O_3$ mixed with 100 SiC). Reaction conditions: 50 mg catalyst, 5000 ppm $NH_3$, 5% $O_2$ balanced in He, gas flow: 100 mL/min, WHSV = 600 mL $NH_3 \cdot h^{-1} \cdot g^{-1}$.

Among these catalysts, $Co_BCu/Al_2O_3$ exhibits the highest activity with the lowest $T_{50}$ value (i.e., the temperature that achieves 50% $NH_3$ conversion) of around 175 °C, with complete $NH_3$ conversion achieved at around 250 °C. The activities of $Co_BCu/Al_2O_3$ and $Co_HCu/Al_2O_3$ are higher than that of $CoCu/Al_2O_3$ (i.e., without reduction) (Fig. 3). Hence, both the addition of Co into CuO and the nature of the reductant affect the catalytic activity. Both $Co_BCu/Al_2O_3$ and $Co_HCu/Al_2O_3$ have higher activities than $CuO/Al_2O_3$, indicating that Co contributes to the activity (Fig. 3). Although $CoO_x/Al_2O_3$ (5 wt% Co loading) is more active than $Co_HCu/Al_2O_3$, the selectivity to $N_2$ is much lower, with the main products comprising NO and $N_2O$ between 250–350 °C (Fig. S10). At 200 °C, $Co_BCu/Al_2O_3$ showed at least a four-fold higher activity than the other catalysts (Fig. 3c). The Co loading was evaluated in the range of 0.1 to 5% for $Co_BCu/Al_2O_3$. As the Co loading increases, the Co species no longer remain as single sites, and the activity of $Co_{1\%}Cu/Al_2O_3$ (Co loading of 1 wt%) and $Co_{5\%}Cu/Al_2O_3$ (Co loading of 5 wt%) are not as high as $Co_BCu/Al_2O_3$ with a Co loading of 0.1 wt% (Fig. S14). Additionally, the selectivity to $N_2$ for $Co_BCu/Al_2O_3$ was > 80%, even at high temperatures, and is lower than $Co_HCu/Al_2O_3$ below 350 °C, but is superior to $Co_HCu/Al_2O_3$ above 350 °C (Fig. 3b, d). Remarkably, despite having the same chemical composition, $Co_BCu/Al_2O_3$ displayed consistently higher activity than $Co_HCu/Al_2O_3$ across all temperature

ranges. At 200 °C, $Co_BCu/Al_2O_3$ achieved nearly 10 times higher $NH_3$ conversion than $Co_HCu/Al_2O_3$ (Fig. 3a). The different electronic structures lead to differences in the strength of the metal−oxygen bonds, which influence both activity and selectivity in the $NH_3$-SCO reaction[38,39]. Catalysts with weaker metal−oxygen bonds tend to exhibit higher rates of reaction, but result in the formation of $NO_x$ (NO and $N_2O$). Compared to $Co_HCu/Al_2O_3$, $Co_BCu/Al_2O_3$ has a lower $1s \rightarrow 4p_z$ antibonding transition and a higher $4p \rightarrow 1s$ bonding transition of Cu-O bonds, indicative of weaker Cu−O bonds[52–54]. Thus, the superior activity and lower selectivity to $N_2$ observed for $Co_BCu/Al_2O_3$ may be attributed to the electronic structure of $Cu^{2+}$ produced by the local coordination environment.

The apparent activation energy of $Co_BCu/Al_2O_3$ is 40.0 kJ/mol, which is lower than that of $Co_HCu/Al_2O_3$ with a value of 63.5 kJ/mol (Fig. 3e). This difference suggests that the superior activity of $Co_BCu/Al_2O_3$ in the $NH_3$-SCO reaction may be attributed to a reduced energy barrier for the reaction. The reaction order for $NH_3$ in the SCO process is 0.4 for $Co_BCu/Al_2O_3$, indicating partial dependence on $NH_3$ concentration, while it is 0 for $Co_HCu/Al_2O_3$ (Fig. 3f), implying that $NH_3$ does not influence the reaction rate under these conditions due to a much higher Cu-O bond energy. The reaction order for $O_2$ also differs between the catalysts, with $Co_BCu/Al_2O_3$ exhibiting an $O_2$ order of 0

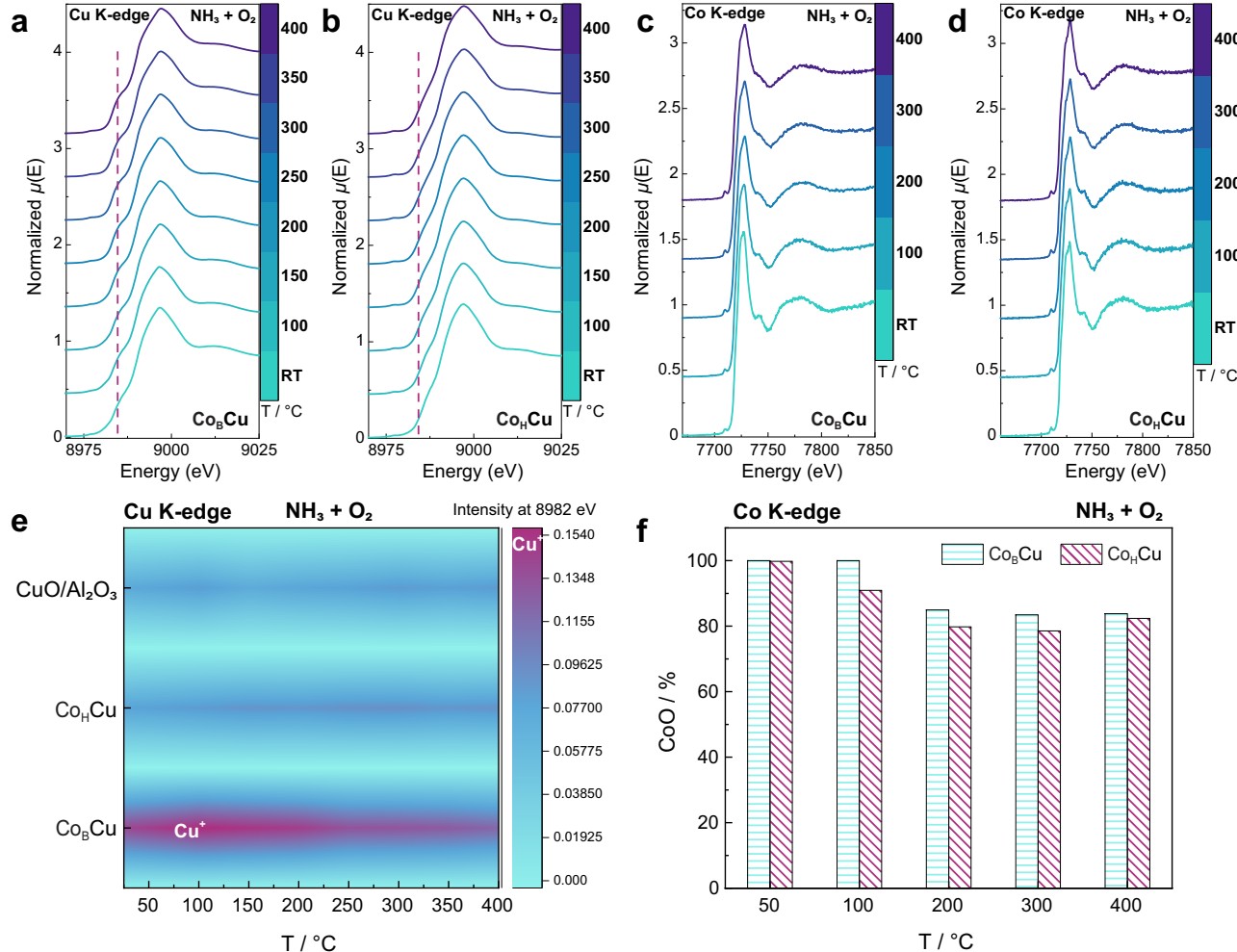

**Fig. 4 | Operando studies of the redox behavior of Cu and Co in Co_BCu/Al_2O_3 and Co_HCu/Al_2O_3.** *Operando* Cu K-edge XAFS of Co_BCu/Al_2O_3 (**a**) and Co_HCu/Al_2O_3 (**b**) as a function of temperature; *operando* Co K-edge XAFS of Co_BCu/Al_2O_3 (**c**) and Co_HCu/Al_2O_3 (**d**); **e** *operando* Cu K-edge XAFS, signal intensity of the Cu$^+$ $1s \rightarrow 4p$ transition peak at 8982 eV in a NH$_3$/O$_2$ atmosphere as a function of temperature; **f** proportion of CoO in Co_BCu/Al_2O_3 and Co_HCu/Al_2O_3 as a function of temperature.

(Fig. 3g), which suggests the participation of lattice oxygen in the reaction. In contrast, Co_HCu/Al_2O_3 shows an O$_2$ order of −0.4 (Fig. 3g), likely due to competitive adsorption between NH$_3$ and O$_2$ at active sites, which inhibits O$_2$ involvement in the reaction. Furthermore, Co_BCu/Al_2O_3 demonstrates good stability under the reaction conditions, showing no significant decrease in catalytic activity or N$_2$ selectivity even after four consecutive reaction cycles. The reduced activity might be caused by the weak metal-support interactions between nanoparticles and support.

**Redox properties of the CuO NPs**

The redox properties of the catalyst play a crucial role in the oxidation of NH$_3$, directly impacting on the overall NH$_3$ oxidation activity[46]. The performance of Cu-based catalysts is related to the Cu$^+$/Cu$^{2+}$ redox couple, and the activity may be correlated with the presence of Cu$^+$, as observed in several Cu-based catalysts[55,56]. Mechanistically, the reaction could be considered to proceed through the reaction of NH$_3$ with Cu$^{2+}$ to form Cu$^I$(NH$_3$)$_2$ as the rate-determining step, followed by the oxidation of Cu$^I$(NH$_3$)$_2$ to Cu$^{2+}$.

To evaluate the redox behavior of the single Co and bulk Cu sites, and monitor structural changes under real NH$_3$-SCO reaction conditions, *operando* XAFS was undertaken in fluorescence mode at the Co K-edge and transmission mode at the Cu K-edge under steady-state conditions at each temperature (Figs. 4, S15 and S16). The formation of reduced Cu$^+$ is considered to be a trigger for the NH$_3$-SCO reaction. In

the Cu K-edge XAFS spectra, the evolution of the Cu$^+$ $1s \rightarrow 4p$ transition peak of Co_BCu/Al_2O_3, Co_HCu/Al_2O_3 and CuO/Al_2O_3 was monitored as a function of temperature during the NH$_3$-SCO reaction (NH$_3$ 5000 ppm, O$_2$ 5%) (Fig. 4e). The Cu$^{2+}$ in Co_BCu/Al_2O_3 is partially reduced to Cu$^+$, even under excess O$_2$, as evidenced by the more pronounced peak intensity at Cu K-edge 8982 eV (the typical feature for Cu$^I$(NH$_3$)$_2$ $1s \rightarrow 4p$, XANES) (Fig. 4e). In contrast, the peak intensity at Cu K-edge 8982 eV in Co_HCu/Al_2O_3 slightly increases, and Cu$^+$ is not detected throughout the entire temperature range in CuO/Al_2O_3 (Fig. 4e, S19 and 20). The redox activity of the single Co sites in Co_BCu/Al_2O_3 and Co_HCu/Al_2O_3 is similar (Fig. 4f). The single Co sites comprise Co$^{2+}$ at room temperature and are partially oxidized to Co$_3$O$_4$ as the temperature increases. Moreover, in different gases from the most reductive, i.e., NH$_3$ without O$_2$, to the most oxidative, i.e., NH$_3$ + O$_2$, the Cu species in Co_BCu/Al_2O_3 are in a more reduced state than Co_HCu/Al_2O_3 (Figs. S17–19), confirming the superior redox ability of the CuO NPs in Co_BCu/Al_2O_3.

As the catalytically active sites are predominantly on the surface, in-situ near ambient pressure near-edge X-ray absorption fine structure (NAP-NEXAFS) was used to investigate the redox properties of the surface Cu species in Co_BCu/Al_2O_3 and Co_HCu/Al_2O_3 (Fig. 5). The redox activity of the Cu species in Co_BCu/Al_2O_3 and Co_HCu/Al_2O_3 was evaluated under NH$_3$ + O$_2$ at different temperatures (Fig. 5a, d). At all temperatures, Co_BCu/Al_2O_3 has a more pronounced peak compared to Co_HCu/Al_2O_3 at 934.2 eV corresponding to Cu$^+$/Cu$^0$, and a broad peak

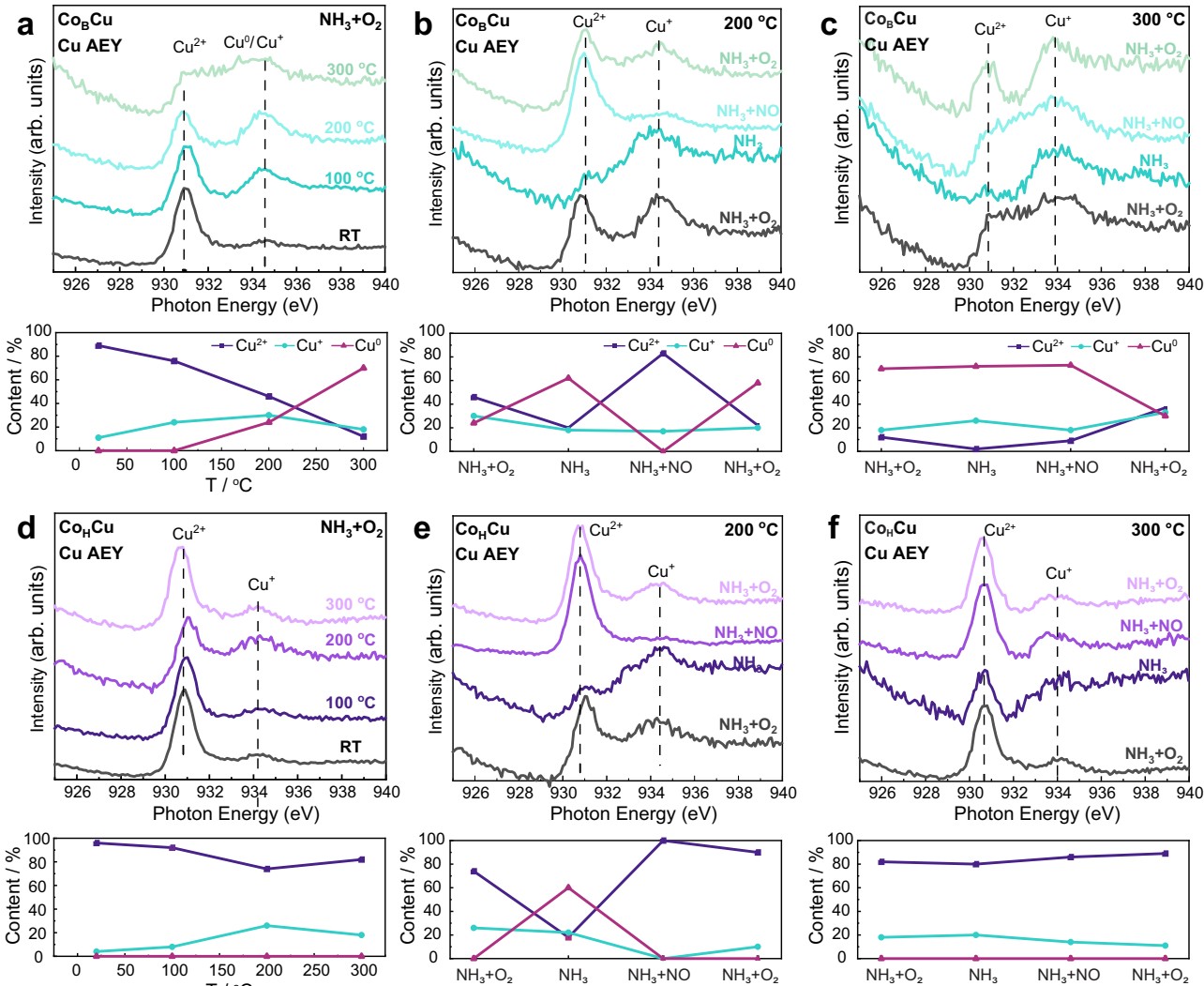

**Fig. 5 | In-situ NAP-NEXAFS spectra and corresponding Cu species distribution of $Co_BCu/Al_2O_3$ and $Co_HCu/Al_2O_3$. a** Cu L-edge (in Auger electron yield (AEY) mode) and corresponding Cu species content of $Co_BCu/Al_2O_3$ under $NH_3 + O_2$ as a function of temperature; Cu L-edge and corresponding Cu species content of $Co_BCu/Al_2O_3$ under various gas atmospheres at 200 °C (**b**) and at 300 °C (**c**) (gas pressure 0.3 mbar); **d** Cu L-edge (AEY mode) and corresponding Cu species content of $Co_HCu/Al_2O_3$ under $NH_3 + O_2$ as a function of temperature; Cu L-edge (AEY mode) and corresponding Cu species content of $Co_HCu/Al_2O_3$ under various gas atmospheres at 200 °C (**e**) and at 300 °C (**f**) (gas pressure 0.3 mbar).

at 930.9 eV corresponding to $Cu^{2+}$. Linear combination fitting (LCF) (see standards in Fig. S20) further confirms that the proportion of reductive Cu species ($Cu^+/Cu^0$) in $Co_BCu/Al_2O_3$ is higher than in $Co_HCu/Al_2O_3$ at all temperatures. Above 200 °C, the reduced Cu species dominate in $Co_BCu/Al_2O_3$ catalyst.

Additionally, NAP-NEXAFS of the $Co_BCu/Al_2O_3$ and $Co_HCu/Al_2O_3$ catalysts under different gas atmospheres further confirmed the superior redox properties of the single Cu sites in $Co_BCu/Al_2O_3$ (Fig. 5). At 200 °C in $NH_3$, the surface Cu species of both $Co_BCu/Al_2O_3$ and $Co_HCu/Al_2O_3$ are nearly completely reduced to $Cu^+/Cu^0$. At 300 °C in $NH_3$, the surface Cu species of $Co_BCu/Al_2O_3$ are nearly completely reduced to $Cu^+/Cu^0$, whereas the surface Cu species of $Co_HCu/Al_2O_3$ are mainly in the form of $Cu^{2+}$. Under $NH_3 + NO$, the surface Cu species in $Co_HCu/Al_2O_3$ are more oxidized than the surface Cu species in $Co_BCu/Al_2O_3$, which become more reduced. Based on the *operando* XAFS and in-situ NAP-NEXAFS studies, the Cu species in the CuO NPs with weaker Cu-O bonds, as in $Co_BCu/Al_2O_3$, exhibit superior redox properties. The longer (and hence weaker) Cu-O bonds in $Co_BCu/Al_2O_3$ result in enhanced redox ability and, consequently, higher activity.

In-Situ DRIFTS confirms different adsorption behavior in $Co_BCu/Al_2O_3$ and $Co_HCu/Al_2O_3$ (Fig. 6)[57]. The bands observed at 1625 and 1256 cm$^{-1}$ may be assigned to asymmetric and symmetric deformation of ammonia chemisorbed on Lewis acid sites of $Al_2O_3$, respectively[23]. The peak at 1405 cm$^{-1}$ may be assigned to $NH_3$ coordinated to $Cu^{58,59}$. The peak at 1460 cm$^{-1}$ originates from Brønsted acid sites on $Al_2O_3$. A peak at 1580 cm$^{-1}$ gradually emerges at temperatures above 250 °C in $Co_BCu/Al_2O_3$, which may be tentatively assigned to nitrate species, with this peak only emerging at temperatures above 300 °C in the presence of $Co_BCu/Al_2O_3$. The unique electronic structure of the CuO NPs in $Co_BCu/Al_2O_3$ which leads to weaker Cu-O bonds, might be an important factor for the enhanced redox activity and the distinct adsorption behavior, which leads to a better catalytic performance, compared to $Co_HCu/Al_2O_3$.

$NH_3$ emissions are expected to rise in the future from mobile vehicles and other industries, as there is increasing interest in using $NH_3$ as a sustainable fuel in particularly for shipping. Consequently, $NH_3$-SCO to $N_2$ is a promising approach to help mitigate $NH_3$ emissions. To improve the performance of earth-abundant metal CoCu-based bimetallic catalysts, single Co sites were doped onto CuO NPs

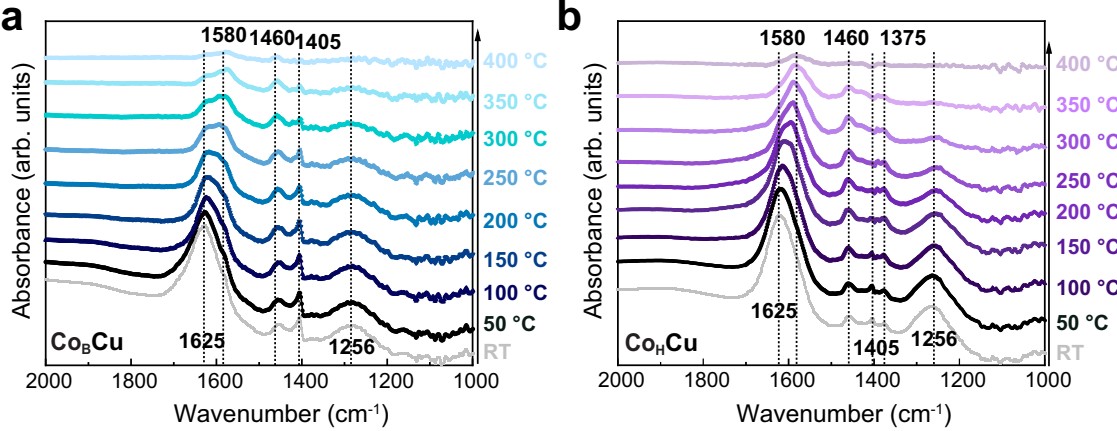

**Fig. 6 | In-Situ DRIFTS spectra.** In-Situ DRIFTS spectra of $Co_BCu/Al_2O_3$ (**a**) and $Cu_HCu/Al_2O_3$ (**b**) as a function of temperature (the catalysts were exposed to a flow of 5000 ppm $NH_3$ and 5% $O_2$ for 20 min at different temperatures.

immobilized on an $Al_2O_3$ support, affording $Co_BCu/Al_2O_3$ and $Co_HCu/Al_2O_3$ catalysts with different electronic structures. Notably, the $Co_BCu/Al_2O_3$ catalyst outperforms the $Co_HCu/Al_2O_3$ catalysts with respect to both activity and selectivity, achieving full conversion of 5000 ppm $NH_3$ at 250 °C, with a selectivity to $N_2 > 80\%$, which is superior to the commercial 1% $Pt/Al_2O_3$ catalyst. The difference in the catalytic performance may be traced to differences in the electronic structure of the CuO NPs in $Co_BCu/Al_2O_3$ and $Co_HCu/Al_2O_3$ using advanced HERFD-XANES and VtC-XES techniques. The lower $1s \rightarrow 4p_z$ transition and higher $1s \rightarrow 4p$ transition ($K_{\beta 2,5}$ feature) of CuO in $Co_BCu/Al_2O_3$ correlate with weaker Cu-O bonds, and these weaker bonds are related to longer Cu-O bonds lengths, as confirmed by STEM, EXAFS and wt-EXAFS. The weaker Cu-O bonds lead to enhanced redox properties of the CuO NPs, and enhanced $NH_3$-SCO activity, further validated by *operando* XAS and in-situ NAP-NEXAFS studies. In contrast to the other catalysts, the oxidation state of the Cu center in $Co_BCu/Al_2O_3$ is able to switch between $Cu^{2+}$ and $Cu^+$ under all gas conditions and at all temperatures studied, which is vital for the catalyst to maintain good performance under real world conditions. Future studies will focus on upscaling and demonstrating the $Co_BCu/Al_2O_3$ catalyst with real fuel gases.

## Methods
### Materials
$Cu(NO_3)_2·3H_2O$, $Co(NO_3)_2·6H_2O$, and $NaBH_4$ were purchased from Sigma Aldrich, and $γ$-$Al_2O_3$ was from Johnson Matthey. All chemicals are used as received.

### Catalyst preparation
**Synthesis of $Co_BCu/Al_2O_3$.** The $Co_BCu/Al_2O_3$ catalyst was prepared via co-reduction of Co and Cu precursors using $NaBH_4$ as the reducing agent, followed by natural oxidation in the air. $γ$-$Al_2O_3$ (0.5 g, Johnson Matthey) was dispersed in $H_2O$ (10 mL) under sonication for 30 min. Solution A containing $Cu(NO_3)_2·3H_2O$ (100 mg) in $H_2O$ (2 mL) and Solution B containing $Co(NO_3)_2·6H_2O$ (2.5 mg) in $H_2O$ (1 mL), were slowly added to the $Al_2O_3$ suspension, avoiding contact with air. After cooling in the ice bath, $NaBH_4$ (78 mg) in $H_2O$ (2 mL) was added, and the resulting suspension was stirred for 30 min under $N_2$. The resulting solid was collected by centrifugation and washed with $H_2O$ (5 ×30 mL), and dried under vacuum, then followed by natural oxidation in the air to afford $Co_BCu/Al_2O_3$.

**Synthesis of $Co_HCu/Al_2O_3$.** The $Co_HCu/Al_2O_3$ catalyst was prepared via co-reduction of Co and Cu precursors under $H_2$, followed by natural oxidation in the air. $γ$-$Al_2O_3$ (0.5 g, Johnson Matthey) was dispersed in

$H_2O$ (10 mL) under sonication for 30 min. Solution A containing $Cu(NO_3)_2·3H_2O$ (100 mg) in $H_2O$ (2 mL) and Solution B containing $Co(NO_3)_2·6H_2O$ (2.5 mg) in $H_2O$ (1 mL), were slowly added to the $Al_2O_3$ suspension. The resulting suspension was heated at 60 °C under stirring until all the solvent had evaporated. The remaining solid was heated to 300 °C for 1 h under 15% $H_2$/Ar at a heating rate of 5 °C/min in a tube furnace, then followed by natural oxidation in the air to afford $Co_HCu/Al_2O_3$.

**Synthesis of $CuO/Al_2O_3$.** $γ$-$Al_2O_3$ (0.5 g, Johnson Matthey) was dispersed in $H_2O$ (10 mL) with vigorous stirring at room temperature. To the resulting suspension, a solution containing $Cu(NO_3)_2·3H_2O$ (100 mg) in $H_2O$ (2 mL) was slowly added. The reaction mixture was then heated at 60 °C under stirring until all the solvent had evaporated. The remaining solid was heated to 300 °C for 1 h under 15% $H_2$/Ar at a heating rate of 5 °C/min in a tube furnace, then followed by natural oxidation in the air to afford $CuO/Al_2O_3$.

**Synthesis of $CoO_x/Al_2O_3$.** $γ$-$Al_2O_3$ (0.5 g, Johnson Matthey) was dispersed in $H_2O$ (10 mL) under sonication for 30 min. Solution containing $Co(NO_3)_2·6H_2O$ (130 mg) in $H_2O$ (2 mL) was slowly added to the $Al_2O_3$ suspension. The resulting suspension was heated at 60 °C under stirring until all the solvent had evaporated. The remaining solid was heated to 300 °C for 1 h under 15% $H_2$/Ar at a heating rate of 5 °C/min in a tube furnace, then followed by natural oxidation in the air to afford $CoO_x/Al_2O_3$.

**Synthesis of $CoCu/Al_2O_3$ (i.e., without reduction).** $γ$-$Al_2O_3$ (0.5 g, Johnson Matthey) was dispersed in $H_2O$ (10 mL) under sonication for 30 min. Solution A containing $Cu(NO_3)_2·3H_2O$ (100 mg) in $H_2O$ (2 mL) and Solution B containing $Co(NO_3)_2·6H_2O$ (2.5 mg) in $H_2O$ (1 mL), were slowly added to the $Al_2O_3$ suspension. The resulting suspension was heated at 60 °C under stirring until all the solvent had evaporated. The remaining solid is $CoCu/Al_2O_3$.

### Ex situ characterization
X-ray diffraction (XRD) patterns were recorded using a StadiP diffractometer (STOE) equipped with a Mo source (Kα = 0.7093165 Å). The instrument operated at an applied voltage of 40 kV and a current of 30 mA. Diffraction data in the 2θ range of 2–40° were systematically collected, with a resolution of 0.015° per step.

X-ray photoelectron spectroscopy (XPS) was conducted using a Thermo Fisher Scientific NEXSA spectrometer. The C $1s$ peak at 284.8 eV served as the standard reference for calibrating the photoelectron energy shift. CasaXPS software was used for data analysis.

Energy dispersive X-ray (EDX) spectroscopy and aberration-corrected bright field (BF) imaging were performed on a JEOL ARM200CF (E01) operating at 200 kV and equipped with JEOL dual silicon drift detectors at the electron Physical Sciences Imaging Centre (ePSIC) at the Diamond Light Source (DLS) (UK). The instrument operated with a convergence semi-angle of 23.0 mrad with BF collection semi-angles of 0–21.9. Single-pass EDX spectra were collected with drift correction. Data were acquired and processed using the Gatan Microscopy Suite (a.k.a. Digital Micrograph)[60]. Samples were prepared via a standard preparation route: a small amount (<20 mg) of catalyst powder was dispersed in approximately 5 ml of ethanol, after sonication and drop casting approximately 1 ml of supernatant onto holey carbon-coated, gold TEM support grids. Gold was used instead of the more typical copper grid to avoid overlapping fluorescent signals with the sample during EDX mapping. The average particle size was calculated based on more than 100 particles for each sample.

HERFD-XANES measurements were conducted at the I20-Scanning beamline at the DLS (UK). The X-ray beam was introduced through Rh-coated optic hutch mirrors, and a Si(111) scanning four-bounce monochromator was employed to select the incident energy. HERFD-XANES spectra were obtained by scanning the incident energy across the range 8800.00 to 9400.00 eV with a step size of 0.15 eV. Samples were homogenized with boron nitride and compressed into pellets with a diameter of 8 mm. XANES analysis was carried out using the Demeter software package.

Extended X-ray absorption fine structure (EXAFS) of the Co K-edge (7709 eV) and Cu K-edge (8979 eV) were carried out at the DLS (UK) and Spring8 (Japan). Samples were directly pressed into pellets for fluorescence measurements of the Co K-edge and transmission measurements of the Cu K-edge. Co foil or Cu foil standards were used for energy shift calibration. XAFS data were merged from 3 spectra to improve signal quality and were processed using the Demeter software package (including Athena and Artemis). Athena software was used to analyze the XANES data. Artemis software was used to fit the $k^2$-weighted EXAFS data in real space with $3.0 \, \text{Å}^{-1} < k < 12.0 \, \text{Å}^{-1}$ and $1.0 \, \text{Å} < R < 3.3 \, \text{Å}$. The calculated amplitude reduction factor, $S_0^2$, from the EXAFS analysis of Cu and Co foil was 0.878 and 0.879, respectively, and were used as fixed parameters for EXAFS fitting. The coordination number and bond length were calculated based on the reported structures from the Crystal open database: Cu (No.9013014), CuO (No. 1011148), Co (No. 9008492), and $Co_3O_4$ (No. 9005898).

### *Operando* Co K-edge and Cu K-edge XAFS

*Operando* XAFS experiments were performed at the Spring8 beamline (Japan). 100 mg of pelletized catalysts were measured at 8780–10200 eV for Cu K edge in transmission mode and 7505–8670 eV for Co K-edge in fluorescence mode at different temperatures and under various gas atmospheres. The pelletized catalyst was exposed to 5000 ppm $NH_3$ and 5% $O_2$ in He (total gas flow rate 100 mL/min) at different temperature from 30 °C to 450 °C. For experiments carried out under various gas atmospheres at 200 °C, the sequence of different gas atmospheres follows $NH_3 + O_2$ (5000 ppm $NH_3$ and 5% $O_2$ in He), $NH_3$ (5000 ppm $NH_3$ in He), $NH_3 + O_2$ (5000 ppm $NH_3$ and 5% $O_2$ in He), and $O_2$ (5% $O_2$ in He). All spectra were recorded under steady-state conditions. Spectra processing was performed with Athena software.

### In-situ DRIFTS

DRIFTS were performed on a PerkinElmer Frontier FT-IR Spectrometer. The sample was heated in He at 350 °C for 30 min to remove surface contamination. After cooling to room temperature, the sample was exposed to 5000 ppm $NH_3$ and 5% $O_2$ in He (total gas flow rate 100 mL/min), for 30 min, during which spectra were recorded. Then, the sample was heated from 30 °C to 450 °C with a temperature ramp of 10 °C/min. The spectra were recorded from 4400 to 500 cm$^{-1}$ with a resolution of 2 cm$^{-1}$. Background spectra were recorded in He and subtracted from the sample spectrum for each measurement.

### In situ near-edge X-ray absorption fine structure (NEXAFS) spectroscopy

In situ NEXAFS experiments were performed on the B07 beamline at the DLS (UK)[61]. The X-ray radiation was sourced from a bending magnet and a plane grating monochromator (PGM) with an energy range from 110 to 2800 eV (soft X-ray range) and flux of >10$^{10}$ photons/s with 0.3 A ring current. The reaction products were monitored online using an electron impact mass spectrometer (Hiden HMT100) connected directly to the prelens chamber. The pressure in the specimen chamber was precisely controlled (HV or 0.1–1 mbar) by simultaneous operation of several mass flow controllers for reactive gases and a PID-controlled throttle valve acting as back pressure controller. Temperature control was provided by two K-type thermocouples. NEXAFS spectra at Cu L-edge (925–940 eV) were measured in Auger electron yield (AEY) mode using a SPECS phoibos 150 hemispherical analyser set.

### Catalytic performance measurements

The performance of the catalysts in the $NH_3$-SCO reaction was evaluated in a fixed-bed flow reactor at a gas flow rate of 100 mL/min, which consists of 5000 ppm $NH_3$, 5% $O_2$, and He balance. Typically, 50 mg of catalyst was placed in the reaction tube, and the quantification of products was performed with an online quadrupole mass spectrometer quantitative gas analyzer (Hiden Analytical, UK). The reactions were investigated at temperatures ranging from 100 to 450 °C and kept stable for at least 30 min after attaining a steady state at each reaction temperature to detect the MS signals of $NH_3$ and $O_2$ and the products, i.e., $N_2$, $N_2O$, and NO. Stability test: 50 mg $Co_BCu/Al_2O_3$ and 100 mg SiC were mixed by grinding and then tested for 4 cycles.

## Data availability

All data generated in this study are provided in the Supplementary Information. Data are available from the corresponding authors upon request.

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

## Acknowledgements

We thank the EPSRC for the UKRI Postdoctoral Fellowship (MSCA) to Lu Chen (EP/X022986/1). We thank the EPSRC (EP/S018204/2, EP/Z001730/1 and EP/Y036220/1) for financial support. We thank the Isaac Newton Trust Grant (24.24(h)), University of Cambridge. We thank the beamline scientists at the SPring-8 (BL14B2) and Diamond Light Source for the provision of beamtimes (I20-Scanning: SP30576-1, B07: SI31867). We thank the scientists in the Electron Physical Science Imaging Centre (EPSIC: MG32996). XAS measurements were performed at BL14B2 of SPring-8 with the approval of the Japan Synchrotron Radiation Research Institute (JASRI) (Proposal No. 2022B1758). We thank Soleil synchrotron (Galaxies beamline 20221122), European Synchrotron Radiation Facility (ESRF) CH6856 and MAX IV (Balder beamline 20230511). We thank the UK Catalysis Hub for the provision of DRIFTS. We thank Dr. Shaoliang Guan (Maxwell Centre, University of Cambridge) for the XPS measurements.

## Author contributions

L.C. and X.G. conceived of the presented idea. L.C. and F.R.W. secured the funding. L.C. carried out catalyst synthesis and catalytic evaluations and X.G. verified the analytical methods. L.C., X.G. and F.R.W. carried out the operando XAS study and L.C. and X.G. analyzed the data. L.C. conducted the XRD measurements and the XPS measurements. HERFD-XANES and VtC-XES studies were supported by S.H. L.C. and X.G. carried out the NAP-NEXAFS studies with support from M.V.S., B.K. and G.H. X.G., C.A. and D.H. conducted the TEM measurements. L.C., X.G. performed DRIFTS experiments with the support of J.C. L.C. and P.J.D. wrote the paper and all authors contributed to the final manuscript.

## Competing interests

The authors declare no competing interests.
