## [Transparent Peer Review file · Nature Communications]

Lowering the Cu-O bond energy in CuO nanocatalysts enhances the efficiency of NH₃ oxidation

Corresponding Author: Professor Paul Dyson

Version 0:

Reviewer comments:

Reviewer #1

(Remarks to the Author)

In this work, the electronic properties of Cu in Cu/Al₂O₃ were tuned by the addition of Co and different reductants. Better NH₃-SCO performance were achieved on CuBCo/Al₂O₃ treated by NaBH₄ rather than H₂. Several operando characterizations were employed to confirm longer and weaker Cu-O bonds on CuBCo/Al₂O₃, contributed to better NH₃-SCO performance. This work is some interesting, but it should be resubmitted after addressed the followed issues.

1. Line 69, Are you sure that "comparatively inexpensive Earth-abundance-based CoO_x-CuO_x/TiO₂ and CoO-68 CuO/Al₂O₃ catalysts result in high selectivity to NO or N₂O"?
2. Line 98, the resolution of EDS mapping should be given in Fig. S5.
3. Line 100-102, the standard card of Al₂O₃ (PDF3 10-0425) should be added in Fig. S7. and the corresponded lattice phase of the diffraction peak should be labeled in Fig. S7. The differences of XRD between CoBCu/Al₂O₃ and CoHCu/Al₂O₃ should be supplemented.
4. CoCu/Al₂O₃ without any reduction should be prepared and employed as the reference, to distinguish the contributions from the interactions between Cu and Co, or the different reductant?
5. Line 151, how about the differences of binding energy of Cu²⁺ 2p 3/2 among the Cu/Al₂O₃, CuBCo and CuHCu?
6. Line 166, It can be seen that the NH₃-SCO activity of Co is lower than CoBCu BUT higher than other three catalysts. What the loading of Co in CoO/Al₂O₃? the enhancement of NH₃-SCO performance on CoBCu is from the addition of Co into Cu/Al₂O₃ or the different reductant or both? Therefore, the NH₃-SCO performance of CuCo/Al₂O₃ prepared without any reductant, as well as its all characterizations should be given out.
7. Line 183, authors thought that "The superior activity of CoBCu/Al₂O₃ may be attributed to the unique electronic structure of Cu²⁺ produced by the local coordination environment", but authors did not explain why "the unique electronic structure of Cu²⁺" would contributed to the better NH₃-SCO performance of CoBCu/Al₂O₃, means that what is relationships between the NH₃-SCO performance (including the catalytic activity and N₂ selectivity) and the electronic structure of Cu²⁺ on the catalysts. The related discussion should be supplemented before Conclusion.
8. Line 298 and 305, Check whether it's CoNCu/Al₂O₃ or CoHCu/Al₂O₃? There is lack the synthesis of Co/Al₂O₃.
9. Line 373, why so high concentration of NH₃ (5000 ppm) was chosen in this work? It is well known that very little NH₃ would be slipped from the upstream NH₃-SCR catalyst, from tens to hundreds ppm, such as ACS Catal. 2023, 13, 6851, and ACS Appl. Mater. Interfaces, 2024, 16, 14694. As well as in the presence of 5-10 VOL. %H₂O and with a high GHSV. Because one factor of low concentration of NH₃, H₂O and a high GHSV would sharply inhibit the NH₃-SCO performance, So the above factors should be considered.

Reviewer #2

(Remarks to the Author)

In this article, Chen et al. synthesized CuO nanoparticles doped with Co atoms and immobilized on Al₂O₃ with two different reduction routes. The Catalyst synthesized via the NaBH₄ reduction route seems to have longer and weaker Cu-O bonds which led to the superior activity for the selective oxidation of NH₃ to N₂, which is a crucial reaction in the context of emission of flue gas and therefore limiting the air pollution to certain extent. The catalysts are characterized by powder XRD, STEM, XAS/XES, XPS, EPR. The topic is very relevant from fundamental aspects and the authors have conducted a very comprehensive study and validated their catalytic test results by in-situ/operando spectroscopy such as XAS, NAP-EXAFS and DRIFTS. I however have few comments and suggestions for improving the manuscript further before accepting in such prestigious journal.

1. The abstract is very unclear to me. The author claimed that the CuO nanoparticles in CoBCu/Al₂O₃ have longer and weaker Cu-O bonds with a lower 1s-4p antibonding transition and higher 4p bonding 1s transition, as demonstrated from scanning transmission electron microscopy, wavelet transform analysis and so on. This sentence is very broad and does not literally conclude anything. For example, microscopy and wavelet transform analysis does not necessarily conclude anything about strength of Cu-O bonds.
2. The free parameters from EXAFS fitting reported in table S1 shows that Co has coordination of approximately 6 which means that Co is in octahedral coordination whereas Cu shows coordination number of around 4 meaning that Cu is in tetrahedral configuration. This is evident from the pre-edge peak at the Cu K-edge XAS spectra. However, my question is whether it is a bimetallic catalyst as the author claimed? Or is it a diluted Co catalyst like single atom alloy? Or it is a Co substituted CuO catalyst? If Co play a significant role in catalysis, then the authors should have observed a sharp change in the Co spectra as Co atoms will reconstruct from octahedral configuration during catalysis to accommodate NH₃.
3. Figure 3 (b) is very difficult to understand. Specially with the legend of CoO/Al₂O₃ and CoHCu/Al₂O₃. If I understand correctly, the CoO/Al₂O₃ showed zero N₂ selectivity in the temperature range between 250-350? Is there any explanation in the text?
4. The Co atoms and their state plays the most important role and is the prime focus of this work. The claim that the Co is atomically dispersed seems to be inclusive. There are a lot of features in figure S09 and S10 at higher R-space (>2 Å) and require further clarifications about those features.
5. Is there any reasons that the authors did not perform any DRIFT study on CoHCu/Al₂O₃ catalyst? If the peak at 1460 cm⁻¹ originates from Brønsted acid sites on Al₂O₃ in CoBCu/Al₂O₃, it could have been useful to observe the same on the CoHCu/Al₂O₃ catalyst.
6. The word "operando" seems to be over interpreted in this work. The authors did not mention any qualitative or quantitative monitoring of the reaction mixture during XAS and DRIFT measurements. At least it is not reported in the experimental section. If the author did not monitor it, they should have clearly stated in the experimental part or in the discussion.

Some minor comments

1. The scale bar in figure 1 (b)-(d) are hardly visible. The author should consider replacing with a brighter color.
2. The authors should consider replacing the dotted lines in figure 1 (e) and (f) by solid lines. Otherwise, it is difficult to see which one is which.
3. The authors did not explain why the powder XRD pattern at figure S7 is very noisy? Are all the samples amorphous? Why reflection from CoO_x or CuO phase are missing?
4. What is the broad peak in the T range of 200-300C of CoBCu/Al₂O₃ during H₂-TPR in fig. S8? I am afraid that there is some presence of bulk CoO_x in this sample.
5. Page 4 above fig. 2, it should be centered in the sentence centered at 4.7 Å⁻¹.

Reviewer #3

(Remarks to the Author)

In this work, the author demonstrates that doping CuO nanoparticles with monodispersed Co atoms using different reductants leads to catalysts with significantly different electronic structures, which in turn affects their NH₃-SCO performance. Various in-situ techniques were employed to characterize the structures of the prepared samples and the reaction pathways. Overall it is a nice piece of work and is potentially suitable for publication in the journal. Several questions need to be addressed:

1. Why does the author believe that the NaBH₄ and H₂ reduction methods induce differences in the electronic structures of the catalysts? A more detailed motivation and literature review should be included in the Introduction.
2. Role of Monodispersed Co Atoms: The "monodispersed" Co atoms appear to play a critical role in enhancing the catalytic activity of CoBCu/Al₂O₃ compared to CuO/Al₂O₃ in Figure 3a. The manuscript should elucidate the specific contributions of cobalt to this improvement. For example, if the Co can help to disperse CuO or change the reaction pathway?
3. The author notes that the selectivity to N₂ for CoBCu/Al₂O₃ is greater than 80% at high temperatures, whereas CoHCu/Al₂O₃ shows 70% selectivity at 450 °C, along with increased production of NO and N₂O (Fig. 3b, d). Given that the T₁₀₀ for fresh CoBCu/Al₂O₃ is approximately 250 °C and for CoHCu/Al₂O₃ is around 350 °C, why is the comparison of N₂ selectivity made at 450 °C? The conclusion of N₂ selectivity between two samples is inaccurate.
4. The T₁₀₀ for NH₃ in CoBCu/Al₂O₃ increases from ~250 °C to ~300 °C, while N₂ selectivity appears to improve from 80-85% to 85-95%. Is this enhancement attributed to the agglomeration of Co species or CuO due to weak metal-support interactions (MSI) with Al₂O₃ prepared by the NaNH₄ reduction method? Does the H₂-reduced CoHCu/Al₂O₃ exhibit similar performance changes?
5. The author mentions that "Operando DRIFTS confirms different adsorption behavior in CoBCu/Al₂O₃ and CoHCu/Al₂O₃ (Fig. 6)." However, Figure 6b presents DRIFTS data for CuO/Al₂O₃, not for CoHCu/Al₂O₃, rendering this description inaccurate.
6. The use of 5000 ppm NO_x without pelleting or dilution sand may lead to hotspot formation in the catalysts during NH₃-SCO reaction. Such hotspots can result in elevated NO_x emissions and reduced cycle stability.

Version 1:

Reviewer comments:

Reviewer #1

(Remarks to the Author)

The authors revised this manuscript appropriately according to the reviewer's comments. However, there are some issues

should be addressed.

1. Authors thought that “Earth-abundant CoOx offers a cost-effective alternative. And the overall activity was enhanced by combined with CuO in catalysts, but leads to the formation of NO or N₂O, so lower N₂ selectivity.” However, the poor N₂ selectivity can be observed on Co/Al₂O₃ from Fig. 3b, Cu/Al₂O₃ displayed the highest N₂ selectivity among the investigated catalysts. Therefore, the N₂ selectivity of Cu-Co catalysts were improved due to the introduction of Cu into Co/Al₂O₃.

2. Authors suggested that “weaker metal–oxygen bonds tend to exhibit higher rates of reaction, but result in the formation of NO_x (NO and N₂O). Compared to CoHCu/Al₂O₃, the superior activity and lower selectivity to N₂ observed for CoBCu/Al₂O₃ may be attributed to the electronic structure of Cu²⁺ produced by the local coordination environment (i.e. weaker Cu–O bonding).” But there was no data on the bond energy of metal-oxygen bonds in the revised manuscript, which should be supplemented.

Reviewer #2

(Remarks to the Author)

The authors have conducted a thorough revision and answered all my questions and incorporated all the suggestions. I do not have any further comments and recommend the article to publish.

The authors should conduct a proper spelling check of the manuscript. For example, caption of Figure 1, instead of HADDF it should be HAADF.

Reviewer #3

(Remarks to the Author)

The authors have well addressed my earlier comments. Acceptance is therefore recommended.

Version 2:

Reviewer comments:

Reviewer #1

(Remarks to the Author)

The authors have addressed my earlier comments. Acceptance is therefore recommended.

Title: Lowering the Cu-O bond energy in CuO nanocatalysts enhances the efficiency of NH₃ oxidation

Note: reviewers' comments appear in **black text**. Our replies appear in **blue text**, and the revised text in the manuscript and supporting information appears in **green**.

Reviewer Comments and point-to-point answers:

Reviewer #1:

In this work, the electronic properties of Cu in Cu/Al₂O₃ were tuned by the addition of Co and different reductants. Better NH₃-SCO performance were achieved on Cu_BCo/Al₂O₃ treated by NaBH₄ rather than H₂. Several operando characterizations were employed to confirm longer and weaker Cu-O bonds on Cu_BCo/Al₂O₃, contributed to better NH₃-SCO performance. This work is some interesting, but it should be resubmitted after addressed the followed issues.

We appreciate that the reviewer describes our work as interesting. Following the reviewer's suggestions (see details below), we have included a more complete activity comparison. Additional experiments have been performed and further discussion has been included. Below are point-to-point responses that address all the reviewer's concerns:

1. Line 69, Are you sure that "comparatively inexpensive Earth-abundance-based CoO_x-CuO_x/TiO₂ and CoO-68 CuO/Al₂O₃ catalysts result in high selectivity to NO or N₂O"?

We apologize for the ambiguity. CuO based catalysts are selective, but the addition of CoO_x leads to formation of NO or N₂O. For the 10CoO-5CuO_x/TiO₂ catalyst we refer to in the manuscript, at 200 °C, the NH₃ removal efficiency is 90 %, with an NO yield of 30 % and N₂O yield of 10 % (Figure 5 in Energy Fuels 2017, 31, 8, 8580–8593, see figure below for review purposes only). For CoO-68 CuO/Al₂O₃ catalysts, "MnO₂ and Co₃O₄ exhibited the highest selectivity to N₂O" (Catalysis Today, 2000, 55, 189–195). We have reworded the original sentence which now reads:

Despite the high activity of noble metal-containing catalysts, their high cost and low selectivity limit practical applications. Earth-abundant CoO_x offers a cost-effective alternative. When combined with CuO in catalysts such as CoO_x-CuO_x/TiO₂ and CoO-CuO/Al₂O₃ catalysts, it enhances the overall activity but leads to the formation of NO or N₂O.^{35,36}

Figure 5. NH₃ removal efficiency (a), NO yield (b), N₂O yield (c), and NO₂ yield (d) over CoO_x-CuO_x/TiO₂ catalysts with different component ratios and the catalytic performance at 170–190 °C (e).

2. Line 98, the resolution of EDS mapping should be given in Fig. S5.

We added a scale bar and the EDS mapping area to Fig. S5.

Fig. S5. Compositional elemental mapping of $\text{Co}_B\text{Cu}/\text{Al}_2\text{O}_3$. **a** STEM image of $\text{Co}_B\text{Cu}/\text{Al}_2\text{O}_3$ (inserted image: EDS mapping area, scale bar: 10 nm); **b-e** EDS elemental mapping of $\text{Co}_B\text{Cu}/\text{Al}_2\text{O}_3$ (scale bar: 10 nm); **f** EDX element distribution pattern. Individual pixel size: 0.5 nm.

3. Line 100-102, the standard card of Al_2O_3 (10-0425) should be added in Fig. S7. and the corresponded lattice phase of the diffraction peak should be labeled in Fig. S7. The differences of XRD between $\text{Co}_B\text{Cu}/\text{Al}_2\text{O}_3$ and $\text{Co}_H\text{Cu}/\text{Al}_2\text{O}_3$ should be supplemented.

Thank you for the suggestion. We have added the standard card of Al_2O_3 to Fig. S7 with the labelled corresponding lattice phase of the diffraction peaks. Peaks at $2\theta = 19.2, 32.0, 36.6, 39.3, 46, 61.5,$ and 67° , are indexed as the (111), (220), (311), (222), (400), (511), and (440) reflections of $\gamma\text{-Al}_2\text{O}_3$. Since the (110), (002), and (111) reflection peaks of CuO are at $2\theta = 32.5, 35.5,$ and 38.7° (CuO: JCPDS #48-1548), they overlap with the (220), (311), and (222) reflections of $\gamma\text{-Al}_2\text{O}_3$. Thus, differences in the XRD patterns of $\text{Co}_B\text{Cu}/\text{Al}_2\text{O}_3$ and $\text{Co}_H\text{Cu}/\text{Al}_2\text{O}_3$ are not observed.

Fig. S7. X-Ray diffraction (XRD) patterns and XPS spectra. **a** XRD patterns of $\text{CoO}_x/\text{Al}_2\text{O}_3$, $\text{CuO}/\text{Al}_2\text{O}_3$, $\text{Co}_B\text{Cu}/\text{Al}_2\text{O}_3$ and $\text{Co}_H\text{Cu}/\text{Al}_2\text{O}_3$. JCPDS card number of $\gamma\text{-Al}_2\text{O}_3$: #10-0425 (Peaks at $2\theta = 19.2, 32.0, 36.6, 39.3, 46, 61.5$ and 67° , are indexed as the (111), (220), (311), (222), (400), (511) and (440) reflections of $\gamma\text{-Al}_2\text{O}_3$.); Cu 2p XPS spectra of $\text{CuO}/\text{Al}_2\text{O}_3$ (**b**) and $\text{CoCu}/\text{Al}_2\text{O}_3$ (**c**).

4. $\text{CoCu}/\text{Al}_2\text{O}_3$ without any reduction should be prepared and employed as the reference, to distinguish the contributions from the interactions between Cu and Co, or the different reductant?

Thank you for the suggestion. We have tested the $\text{CoCu}/\text{Al}_2\text{O}_3$ catalysts without any reduction in the $\text{NH}_3\text{-SCO}$ reaction as a reference, see revised Figure 3a&b. Both the addition of Co into CuO and the nature of the reductant effect the catalytic activity. The synthetic procedure has been added to the Experimental part and the following discussion added.

Among these catalysts, $\text{Co}_B\text{Cu}/\text{Al}_2\text{O}_3$ exhibits the highest activity with the lowest T_{50} value (i.e. the temperature that achieves 50% NH_3 conversion) of around 175°C , with complete NH_3 conversion achieved at around 250°C . The activities of $\text{Co}_B\text{Cu}/\text{Al}_2\text{O}_3$ and $\text{Co}_H\text{Cu}/\text{Al}_2\text{O}_3$ are higher than that of $\text{CoCu}/\text{Al}_2\text{O}_3$ (i.e. without reduction) (Fig. 3). Hence, both the addition of Co into CuO and the nature of the reductant effect the catalytic activity.

Figure 3 Evaluation of Co_BCu/Al₂O₃, Co_HCu/Al₂O₃ and control catalysts in the NH₃-SCO reaction. NH₃ conversion (a) and selectivity to N₂ (b) as a function of temperature (note that CoCu/Al₂O₃ corresponds to the catalyst without reduction); Product distribution at 200 °C (c) and 450 °C (d); e Activation energy for the Co_BCu/Al₂O₃ and Co_HCu/Al₂O₃ catalyzed reaction; Reaction order for NH₃ (f) and O₂ (g); Stability tests of Co_BCu/Al₂O₃ over 4 cycles (h, i) (50 mg Co_BCu/Al₂O₃ mixed with 100 SiC). Reaction conditions: 50 mg catalyst, 5000 ppm NH₃, 5% O₂ balanced in He, gas flow: 100 mL/min, WHSV = 600 mL NH₃·h⁻¹·g⁻¹.

Synthesis of CoCu/Al₂O₃ (i.e. without reduction): γ-Al₂O₃ (0.5 g, Johnson Matthey) was dispersed in H₂O (10 mL) under sonication for 30 min. Solution A containing Cu(NO₃)₂·3H₂O (100 mg) in H₂O (2 mL) and Solution B containing Co(NO₃)₂·6H₂O (2.5 mg) in H₂O (1 mL), were slowly added to the Al₂O₃ suspension. The resulting suspension was heated at 60 °C under stirring until all the solvent had evaporated. The remaining solid is CoCu/Al₂O₃.

5. Line 151, how about the differences of binding energy of Cu²⁺ 2p_{3/2} among the Cu/Al₂O₃, Cu_BCo and Cu_HCo?

The binding energy of Cu²⁺ 2p_{3/2} in CuO/Al₂O₃ is 934.6 eV, which is similar to that in Cu_HCo (934.7 eV), and higher than in Cu_BCo (933.5 eV). These values are consistent with the VtC-XES and HERFD-XANES, and indicate that the electronic state of Cu in Cu_HCo is similar to CuO/Al₂O₃, but different to Cu_BCo. Thank you for point this out, it strengthens our conclusions.

Additionally, XPS of $\text{Co}_\text{B}\text{Cu}/\text{Al}_2\text{O}_3$, $\text{Co}_\text{H}\text{Cu}/\text{Al}_2\text{O}_3$, $\text{CuO}/\text{Al}_2\text{O}_3$ and $\text{CoCu}/\text{Al}_2\text{O}_3$ exhibit $\text{Cu}^{2+} 2\text{p}_{3/2}$ peaks at 933.5, 934.7, 934.6 and 934.7 eV, respectively (Fig. 2d, e, Fig. S7b, c). Thus, the electronic states of Cu in $\text{Cu}_\text{H}\text{Co}$, $\text{CuO}/\text{Al}_2\text{O}_3$ and $\text{CoCu}/\text{Al}_2\text{O}_3$ are similar, with that of $\text{Cu}_\text{B}\text{Co}$ being lower (Fig. S7b). The combination of HERFD-XANES, VtC-XES and XPS suggests that the Cu-O bonds in the $\text{Co}_\text{B}\text{Cu}/\text{Al}_2\text{O}_3$ catalyst are weakened relative to those observed in $\text{Co}_\text{H}\text{Cu}/\text{Al}_2\text{O}_3$ (Fig. 2a), which is as expected as the Cu-O bonds in $\text{Co}_\text{B}\text{Cu}/\text{Al}_2\text{O}_3$ are longer (Fig. 1g, h). This feature indicates that the Cu-O bonds in $\text{Co}_\text{B}\text{Cu}/\text{Al}_2\text{O}_3$ will be more reactive toward NH_3 , leading to reduction of Cu^{2+} and concomitant oxidation of NH_3 .

6. Line 166, It can be seen that the NH_3 -SCO activity of Co is lower than $\text{Co}_\text{B}\text{Cu}$ BUT higher than other three catalysts. What the loading of Co in $\text{CoO}/\text{Al}_2\text{O}_3$? the enhancement of NH_3 -SCO performance on $\text{Co}_\text{B}\text{Cu}$ is from the addition of Co into $\text{Cu}/\text{Al}_2\text{O}_3$ or the different reductant or both? Therefore, the NH_3 -SCO performance of $\text{CuCo}/\text{Al}_2\text{O}_3$ prepared without any reductant, as well as its all characterizations should be given out.

The loading of Co in $\text{CoO}/\text{Al}_2\text{O}_3$ is 5% and the synthetic procedure has been added. Both the addition of Co and the different reductants have enhanced the catalytic performance as discussed in Point 4 (see above), and the characterization of $\text{CuCo}/\text{Al}_2\text{O}_3$ has been included in the Supporting Information.

Among these catalysts, $\text{Co}_\text{B}\text{Cu}/\text{Al}_2\text{O}_3$ exhibits the highest activity with the lowest T_{50} value (i.e. the temperature that achieves 50% NH_3 conversion) of around 175 °C, with complete NH_3 conversion achieved at around 250 °C. The activities of $\text{Co}_\text{B}\text{Cu}/\text{Al}_2\text{O}_3$ and $\text{Co}_\text{H}\text{Cu}/\text{Al}_2\text{O}_3$ are higher than that of $\text{CoCu}/\text{Al}_2\text{O}_3$ (i.e. without reduction) (Fig. 3). Hence, both the addition of Co into CuO and the nature of the reductant effect the catalytic activity.

Synthesis of $\text{Co}_\text{x}/\text{Al}_2\text{O}_3$: $\gamma\text{-Al}_2\text{O}_3$ (0.5 g, Johnson Matthey) was dispersed in H_2O (10 mL) under sonication for 30 min. Solution containing $\text{Co}(\text{NO}_3)_2 \cdot 6\text{H}_2\text{O}$ (130 mg) in H_2O (2 mL) was slowly added to the Al_2O_3 suspension. The resulting suspension was heated at 60 °C under stirring until all the solvent had evaporated. The remaining solid was heated to 300 °C for 1 h under 15% H_2/Ar at a heating rate of 5 °C/min in a tube furnace, then followed by naturally oxidation in the air to afford $\text{CoO}_\text{x}/\text{Al}_2\text{O}_3$.

7. Line 183, authors thought that "The superior activity of $\text{Co}_\text{B}\text{Cu}/\text{Al}_2\text{O}_3$ may be attributed to the unique electronic structure of Cu^{2+} produced by the local coordination environment ", but authors did not explain why "the unique electronic structure of Cu^{2+} " would contributed to the better NH_3 -SCO performance of $\text{Co}_\text{B}\text{Cu}/\text{Al}_2\text{O}_3$, means that what is relationships between the NH_3 -SCO performance (including the catalytic activity and N_2 selectivity) and the electronic structure of Cu^{2+} on the catalysts. The related discussion should

be supplemented before Conclusion.

Thank you for the suggestion regarding the relationship between the electronic structure of Cu^{2+} and the NH_3 -SCO performance. Compared with $\text{Co}_\text{H}\text{Cu}/\text{Al}_2\text{O}_3$, $\text{Co}_\text{B}\text{Cu}/\text{Al}_2\text{O}_3$ shows higher activity but lower N_2 selectivity, which is consistent with previous findings, i.e. catalysts with weaker metal–oxygen bonds exhibit higher rates of reaction but lower N_2 selectivity. We have reworded the related discussion to improve clarity.

The different electronic structures leads to differences in the strength of the metal–oxygen bonds, which influences both activity and selectivity in the NH_3 -SCO reaction.^{40,41} Catalysts with weaker metal–oxygen bonds tend to exhibit higher rates of reaction, but result in the formation of NO_x (NO and N_2O). Compared to $\text{Co}_\text{H}\text{Cu}/\text{Al}_2\text{O}_3$, the superior activity and lower selectivity to N_2 observed for $\text{Co}_\text{B}\text{Cu}/\text{Al}_2\text{O}_3$ may be attributed to the electronic structure of Cu^{2+} produced by the local coordination environment (i.e. weaker Cu – O bonding).

8. Line 298 and 305, Check whether it's $\text{Co}_\text{N}\text{Cu}/\text{Al}_2\text{O}_3$ or $\text{Co}_\text{H}\text{Cu}/\text{Al}_2\text{O}_3$? There is lack the synthesis of $\text{Co}/\text{Al}_2\text{O}_3$.

Thank you for noticing these inconsistencies. We have carefully checked and corrected the formula, and added the synthesis procedure of $\text{CoO}_x/\text{Al}_2\text{O}_3$.

Synthesis of $\text{Co}_x/\text{Al}_2\text{O}_3$: γ - Al_2O_3 (0.5 g, Johnson Matthey) was dispersed in H_2O (10 mL) under sonication for 30 min. Solution containing $\text{Co}(\text{NO}_3)_2 \cdot 6\text{H}_2\text{O}$ (130 mg) in H_2O (2 mL) was slowly added to the Al_2O_3 suspension. The resulting suspension was heated at 60 °C under stirring until all the solvent had evaporated. The remaining solid was heated to 300 °C for 1 h under 15% H_2/Ar at a heating rate of 5 °C/min in a tube furnace, then followed by naturally oxidation in the air to afford $\text{CoO}_x/\text{Al}_2\text{O}_3$.

9. Line 373, why so high concentration of NH_3 (5000 ppm) was chosen in this work? It is well known that very little NH_3 would be slipped from the upstream NH_3 -SCR catalyst, from tens to hundreds ppm, such as ACS Catal.2023,13,6851, and ACS Appl. Mater. Interfaces, 2024,16,14694. As well as in the presence of 5-10 VOL. % H_2O and with a high GHSV. Because one factor of low concentration of NH_3 , H_2O and a high GHSV would sharply inhibit the NH_3 -SCO performance, So the above factors should be considered.

Thank you for raising this issue. For the SCO in mobile applications, the NH_3 concentration is indeed from tens to hundreds ppm. We used a high concentration of NH_3 (5000 ppm) which is characteristic of the NH_3 concentration in waste gas streams from industrial plants, for example raw coke oven gas contains 6-8 g/Nm^3 of NH_3 before scrubbing, melamine and nitric acid production contains 6-8 g/Nm^3 of NH_3 . Also, the concentration we used aligns with the gas feed used by other groups (*Nat. Catal.*, 2019, 2, 157–163, *ACS Catal.*, 2021, 11, 2141–2149, *Chem. Commun.*, 2003, 1280–1281).

The gas hourly space velocity (GHSV) we used is $100,000 \text{ h}^{-1}$, similar with work published in *ACS Catal.* 2023,13,6851 ($200,000 \text{ h}^{-1}$) and *ACS Appl. Mater. Interfaces*, 2024,16,14694 ($100,000 \text{ h}^{-1}$). We have added the following statement to the manuscript:

5000 ppm NH_3 and GHSV $100,000 \text{ h}^{-1}$ are used as the stander test conditions, that are consistent with waste gas streams in industrial processes and have been used in previous studies.⁵⁰⁻⁵³

Reviewer #2:

In this article, Chen et al. synthesized CuO nanoparticles doped with Co atoms and immobilized on Al₂O₃ with two different reduction routes. The Catalyst synthesized via the NaBH₄ reduction route seems to have longer and weaker Cu-O bonds which led to the superior activity for the selective oxidation of NH₃ to N₂, which is a crucial reaction in the context of emission of flue gas and therefore limiting the air pollution to certain extent. The catalysts are characterized by powder XRD, STEM, XAS/XES, XPS, EPR. The topic is very relevant from fundamental aspects and the authors have conducted a very comprehensive study and validated their catalytic test results by in-situ/operando spectroscopy such as XAS, NAP-EXAFS and DRIFTS. I however have few comments and suggestions for improving the manuscript further before accepting in such prestigious journal.

We appreciate the reviewer's insightful evaluation, comprehensive and constructive feedback which serves to improve the quality of our work. We have taken into account each point raised, incorporated new results and discussions accordingly. We believe our efforts will effectively address the reviewer's concerns.

1. The abstract is very unclear to me. The author claimed that the the CuO nanoparticles in Co_BCu/Al₂O₃ have longer and weaker Cu-O bonds with a lower 1s→4p_z antibonding transition and higher 4p bonding→1s transition, as demonstrated from scanning transmission electron microscopy, wavelet transform analysis and so on. This sentence is very broad and does not literally conclude anything. For example, microscopy and wavelet transform analysis does not necessarily conclude anything about strength of Cu-O bonds.

Thank you for raising this issue. We have updated our abstract to provide greater clarity, specifically regarding the link between Cu-O bond lengths (evidenced from STEM, EXAFS fitting and WT-EXAFS analysis) and bond strengths (evidenced from XPS, HERFD-XANES, and VtC-XES), the consequent effect on redox ability (evidenced from *operando* XAFS and *in-situ* NAP-NEXAFS), and the resulting impact on catalytic activity. The revised abstract is provided below:

Compared to Co_HCu/Al₂O₃, the CuO nanoparticles in Co_BCu/Al₂O₃ have longer Cu-O bonds (demonstrated from scanning transmission electron microscopy, wavelet transform analysis of the extended X-ray absorption fine structure spectra). The longer Cu-O bonds in Co_BCu/Al₂O₃ are weaker than those in Co_HCu/Al₂O₃, with a lower 1s → 4p_z antibonding transition and higher 4p bonding→1s transition (as demonstrated from high energy resolution fluorescence detected - X-ray absorption near edge structure and valence-to-core X-ray emission spectroscopy). The weaker Cu-O bonds in Co_BCu/Al₂O₃ lead to superior redox activity of the CuO nanoparticles, evidenced from *operando* X-ray absorption fine structure and *in-situ* near ambient pressure-near edge X-ray absorption fine structures studies. Such superior redox

properties of CuO in Co_BCu/Al₂O₃ results in a much reduced activation energy of Co_BCu/Al₂O₃ compared to Co_HCu/Al₂O₃ (40.0 vs. 63.5 kJ/mol), thus leading to an enhancement in catalytic performance in the selective catalytic oxidation of NH₃ to N₂, a critical reaction used to eliminate NH₃ emissions in flue gases.

2. The free parameters from EXAFS fitting reported in table S1 shows that Co has coordination of approximately 6 which means that Co is in octahedral coordination whereas Cu shows coordination number of around 4 meaning that Cu is in tetrahedral configuration. This is evident from the pre-edge peak at the Cu K-edge XAS spectra. However, my question is whether it is a bimetallic catalyst as the author claimed? Or is it a diluted Co catalyst like single atom alloy? Or it is a Co substituted CuO catalyst? If Co play a significant role in catalysis, then the authors should have observed a sharp change in the Co spectra as Co atoms will reconstruct from octahedral configuration during catalysis to accommodate NH₃.

Based on the EXAFS fitting results, no Co-Co bonds are observed, indicating that Co atoms are monodispersed (i.e., present as single Co atoms) in the CuO nanoparticles. Since both CuO and Co species are in the oxidized state, Co_BCu/Al₂O₃ is not well-described as an alloy, thus we classify it a bimetallic catalyst containing monodispersed Co atoms. Three techniques (including STEM, EXAFS fitting and WT-EXAFS analysis) consistently demonstrate that Cu-O bond length in Co_BCu/Al₂O₃ (1.96 Å ± 0.005) is longer than in Co_HCu/Al₂O₃ (1.93 Å ± 0.003). Given the Co-O bond length in Co_BCu/Al₂O₃ is longer (2.04 Å ± 0.0002) than Cu-O bonds, we propose that Co substitution into the CuO lattice elongates the Cu-O bonds. This bond lengthening reduces the Cu-O bond dissociation energy (as evidenced by XPS, HERFD-XANES, and VtC-XES), enhances the redox ability of CuO (supported by operando XAFS and in-situ NAP-NEXAFS), and ultimately improves catalytic activity. The absence of sharp changes in the Co spectra suggests that Cu species are likely the active sites in NH₃-SCO.

3. Figure 3 (b) is very difficult to understand. Specially with the legend of CoO/Al₂O₃ and Co_HCu/Al₂O₃. If I understand correctly, the CoO/Al₂O₃ showed zero N₂ selectivity in the temperature range between 250-350? Is there any explanation in the text?

Your understanding is correct, the CoO_x/Al₂O₃ showed zero N₂ selectivity between 250-350 °C. The main products are NO and N₂O in this temperature range, the detailed selectivity is shown in Fig. S13. CoO_x has the highest activity in NH₃ oxidation among non-noble metals but suffers from over-oxidation to NO and N₂O (4 NH₃ + 5O₂ → 4 NO + 6 H₂O and 2 NH₃ + 2 O₂ → N₂O + 3 H₂O) as reported (ACS Catal. 2023, 13, 13816–13827, Applied Catalysis A: General 1998, 175, 147-157). We have modified the discussion to improve clarity and avoid the confusion caused by our original wording.

Both $\text{Co}_B\text{Cu}/\text{Al}_2\text{O}_3$ and $\text{Co}_H\text{Cu}/\text{Al}_2\text{O}_3$ have higher activities than $\text{CuO}/\text{Al}_2\text{O}_3$, indicating that Co contributes to the activity (Fig. 3). Although $\text{CoO}_x/\text{Al}_2\text{O}_3$ (5 wt% Co loading) is more active than $\text{Co}_H\text{Cu}/\text{Al}_2\text{O}_3$, the selectivity to N_2 is much lower, with the main products comprising NO and N_2O between 250-350 °C (Fig. S10).

4. The Co atoms and their state plays the most important role and is the prime focus of this work. The claim that the Co is atomically dispersed seems to be inclusive. There are a lot of features in figure S09 and S10 at higher R-space ($>2 \text{ \AA}$) and require further clarifications about those features.

Thank you for the suggestion. Due to the low loading of Co (0.1 wt%), the quality of Co K-edge EXAFS data is low, and the R-factor is more than 0.02 if the second shell is considered. Given the low Co loading and limited signal-to-noise ratio, these features might originate from background fluctuation or minor contributions from multiple scattering paths. Therefore, no distinct second-shell peaks attributable to Co–Co were fitted. Moreover, EXAFS fitting cannot distinguish between Co–O–Cu and Co–O–Co bonds. However, the EXAFS fitting and STEM-EDX results reveal Co–O bonds and a uniform Co distribution, allowing us to conclude that Co is atomically dispersed.

5. Is there any reasons that the authors did not perform any DRIFT study on $\text{Co}_H\text{Cu}/\text{Al}_2\text{O}_3$ catalyst? If the peak at 1460 cm^{-1} originates from Brønsted acid sites on Al_2O_3 in $\text{Co}_B\text{Cu}/\text{Al}_2\text{O}_3$, it could have been useful to observe the same on the $\text{Co}_H\text{Cu}/\text{Al}_2\text{O}_3$ catalyst.

A DRIFTS study of $\text{Co}_H\text{Cu}/\text{Al}_2\text{O}_3$ was conducted and the data has been added to Fig 6. The peak at 1460 cm^{-1} originates from Brønsted acid sites on Al_2O_3 in $\text{Co}_B\text{Cu}/\text{Al}_2\text{O}_3$, it also observed on the $\text{Co}_H\text{Cu}/\text{Al}_2\text{O}_3$ catalyst. We have added the following discussion to the manuscript:

In-situ DRIFTS confirms different adsorption behaviour in $\text{Co}_B\text{Cu}/\text{Al}_2\text{O}_3$ and $\text{Co}_H\text{Cu}/\text{Al}_2\text{O}_3$ (Fig. 6).⁴⁸ The peak at 1460 cm^{-1} originates from Brønsted acid sites on Al_2O_3 . A peak at 1580 cm^{-1} gradually emerges at temperatures above 250 °C in $\text{Co}_B\text{Cu}/\text{Al}_2\text{O}_3$, which may be tentatively assigned to nitrate species, with this peak only emerging at temperatures above 300 °C in the presence of $\text{Co}_B\text{Cu}/\text{Al}_2\text{O}_3$.

Figure 6 *In-situ* DRIFTS spectra. *In-situ* DRIFTS spectra of $\text{Co}_B\text{Cu}/\text{Al}_2\text{O}_3$ (a) and $\text{Cu}_H\text{Cu}/\text{Al}_2\text{O}_3$ (b) as a function of temperature (the catalysts were exposed to a flow of 5000 ppm NH_3 and 5% O_2 for 20 min at different temperatures).

6. The word “operando” seems to be over interpreted in this work. The authors did not mention any qualitative or quantitative monitoring of the reaction mixture during XAS and DRIFT measurements. At least it is not reported in the experimental section. If the author did not monitor it, they should have clearly stated in the experimental part or in the discussion.

XAS and DRIFT experiments were performed in 5000 ppm NH_3 and 5% O_2 in He (total gas flow rate 100 mL/min), the experiment conditions are exactly the same in the catalytic tests. We used mass spectrometry for qualitative monitoring, but not quantitative measurements, so we changed ‘Operando’ to ‘*In-situ*’ for DRIFTS. We have provided further details for clarity:

Operando Co K-edge and Cu K-edge XAFS

Operando XAFS experiments were performed at the Spring8 beamline (Japan). 100 mg of pelletized catalysts were measured at 8780-10200 eV for Cu K edge in transmission mode and 7505-8670 eV for Co K-edge in fluorescence mode at different temperatures and under various gas atmospheres. The pelletized catalyst was exposed to 5000 ppm NH_3 and 5% O_2 in He (total gas flow rate 100 mL/min) at different temperature from 30 °C to 450 °C. For experiments carried out under various gas atmospheres at 200 °C, the sequence of different gas atmospheres follows NH_3+O_2 (5000 ppm NH_3 and 5% O_2 in He), NH_3 (5000 ppm NH_3 in He), NH_3+O_2 (5000 ppm NH_3 and 5% O_2 in He) and O_2 (5% O_2 in He). All spectra were recorded under steady-state conditions. Spectra processing was performed with Athena software.

***In-situ* DRIFTS**

DRIFTS were performed on a PerkinElmer Frontier FT-IR Spectrometer. The sample was heated in He at 350 °C for 30 min to remove surface contamination. After cooling to room temperature, the sample was exposed to 5000 ppm NH_3 and 5% O_2 in He (total gas flow rate 100 mL/min), for 30 minutes, during which spectra were recorded. Then, the sample was heated

from 30 °C to 450 °C with a temperature ramp of 10 °C/min. The spectra were recorded from 4400 to 500 cm^{-1} with a resolution of 2 cm^{-1} . Background spectra were recorded in He and subtracted from the sample spectrum for each measurement.

Some minor comments

Thank you so much for the careful review, we have checked the manuscript and have hopefully fixed all the formatting errors.

1. The scale bar in figure 1(b)-(d) are hardly visible. The author should consider replacing with a brighter color.

We improved the visibility of the scale bar in Figure 1(b)-(d) as recommended.

Figure 1 Characterization of $\text{Co}_B\text{Cu}/\text{Al}_2\text{O}_3$ and $\text{Co}_H\text{Cu}/\text{Al}_2\text{O}_3$. BF images (a) and HADDF image (b) of $\text{Co}_B\text{Cu}/\text{Al}_2\text{O}_3$ (Scale bar = 2 nm); BF image (c) and HADDF image (d) of $\text{Co}_H\text{Cu}/\text{Al}_2\text{O}_3$ (Scale bar = 2 nm); Cu K-edge HERFD-XANES (e) and Co K-edge XANES (f) of $\text{Co}_B\text{Cu}/\text{Al}_2\text{O}_3$ and $\text{Co}_H\text{Cu}/\text{Al}_2\text{O}_3$; 2D WT-EXAFS maps of the CuO NPs in $\text{Co}_B\text{Cu}/\text{Al}_2\text{O}_3$ (g) and $\text{Co}_H\text{Cu}/\text{Al}_2\text{O}_3$ (h).

2. The authors should consider replacing the dotted lines in figure 1 (e) and (f) by solid lines. Otherwise, it is difficult to see which one is which.

We replaced the dotted lines with solid lines in Figure 1(e) and (f) as recommended, please see the above figure.

3. The authors did not explain why the powder XRD pattern at figure S7 is very noisy? Are all the samples amorphous? Why reflection from CoO_x or CuO phase are missing?

Thank you for the suggestion. We have added the standard card of Al_2O_3 to Fig. S7 with the labelled corresponding lattice phase of the diffraction peaks. Peaks at $2\theta = 19.2, 32.0, 36.6, 39.3, 46, 61.5,$ and 67° , are indexed as the (111), (220), (311), (222), (400), (511), and (440) reflections of $\gamma\text{-Al}_2\text{O}_3$. Since the (110), (002), and (111) reflection peaks of CuO are at $2\theta = 32.5, 35.5,$ and 38.7° (CuO : JCPDS #48-1548), they overlap with the (220), (311), and (222) reflections of $\gamma\text{-Al}_2\text{O}_3$. Thus, differences in the XRD patterns of $\text{Co}_B\text{Cu}/\text{Al}_2\text{O}_3$

and $\text{Co}_\text{H}\text{Cu}/\text{Al}_2\text{O}_3$ are not observed.

Fig. S7. X-Ray diffraction (XRD) patterns and XPS spectra. **a** XRD patterns of $\text{CoO}_x/\text{Al}_2\text{O}_3$, $\text{CuO}/\text{Al}_2\text{O}_3$, $\text{Co}_\text{B}\text{Cu}/\text{Al}_2\text{O}_3$ and $\text{Co}_\text{H}\text{Cu}/\text{Al}_2\text{O}_3$. JCPDS card number of $\gamma\text{-Al}_2\text{O}_3$: #10-0425 (Peaks at $2\theta = 19.2, 32.0, 36.6, 39.3, 46, 61.5$ and 67° , are indexed as the (111), (220), (311), (222), (400), (511) and (440) reflections of $\gamma\text{-Al}_2\text{O}_3$.); $\text{Cu } 2\text{p}$ XPS spectra of $\text{CuO}/\text{Al}_2\text{O}_3$ (**b**) and $\text{CoCu}/\text{Al}_2\text{O}_3$ (**c**).

4. What is the broad peak in the T range of 200-300C of $\text{Co}_\text{B}\text{Cu}/\text{Al}_2\text{O}_3$ during H_2 -TPR in fig. S8? I am afraid that there is some presence of bulk CoO_x in this sample.

The broad peak in the T range 200-300 $^\circ\text{C}$ in H_2 -TPR of $\text{Co}_\text{B}\text{Cu}/\text{Al}_2\text{O}_3$ may originate from bulk CoO_x , due to the aggregation of Co under H_2 . The Co atoms in $\text{Co}_\text{B}\text{Cu}/\text{Al}_2\text{O}_3$ under the oxidative reaction conditions maintain a monodispersed state, as evidenced from the operando XAFS.

H_2 -temperature programmed reduction (TPR) confirms that incorporation of Co atoms in $\text{Co}_\text{B}\text{Cu}/\text{Al}_2\text{O}_3$ shifts the reduction temperature of CuO NPs by 34 $^\circ\text{C}$ (first peak) to lower temperature (Fig. S8), indicating that Co promotes the reduction of CuO , probably *via* the asymmetric Co-O-Cu bond.⁴⁵ In contrast, for $\text{Co}_\text{H}\text{Cu}/\text{Al}_2\text{O}_3$, two separate peaks indicate weaker interactions between the Cu and Co species. The broad peak in the T range of 200-300 $^\circ\text{C}$ of $\text{Co}_\text{B}\text{Cu}/\text{Al}_2\text{O}_3$ in H_2 -TPR may originate from bulk CoO_x , due to the aggregation of Co under H_2 .

5. Page 4 above fig. 2, it should be centered in the sentence cantered at 4.7 \AA^{-1} .

Thank you for spotting the error, we have corrected 'cantered' to 'centered'.

Reviewer #3:

In this work, the author demonstrates that doping CuO nanoparticles with monodispersed Co atoms using different reductants leads to catalysts with significantly different electronic structures, which in turn affects their NH₃-SCO performance. Various in-situ techniques were employed to characterize the structures of the prepared samples and the reaction pathways. Overall it is a nice piece of work and is potentially suitable for publication in the journal. Several questions need to be addressed:

We appreciate the reviewer's positive feedback and excellent suggestion to include a more complete literature review and more reasonable activity comparison, and to improve the catalytic stability through avoiding hot spots. Additional discussion and experiments and updated figures have been included.

1. Why does the author believe that the NaBH₄ and H₂ reduction methods induce differences in the electronic structures of the catalysts? A more detailed motivation and literature review should be included in the Introduction.

We understand the concerns from the reviewer about the reduction methods inducing differences in the electronic structures, and We propose that reduction with a hydride source (NaBH₄) versus a neutral hydrogen source (H₂) may lead to distinct electronic structures. A detailed motivation and literature review have been added to the Introduction:

The activity and the selectivity of bimetallic catalysts are significantly influenced by their geometric and electronic structures.³⁷⁻³⁹ The different electronic structures leads to differences in the strength of the metal–oxygen bond, which determines the distribution of ammonia oxidation products.^{40,41} Oxides with high metal–oxygen bond strengths exhibit lower rates of reaction and facilitate a high selectivity to N₂. In contrast, metal oxides with weak metal–oxygen bond strengths lead to the formation of NO_x (NO and N₂O). The unique structural features of single site doped bimetallic NPs allows their electronic properties to be tuned more precisely than their monometallic counterparts, which provides a facile approach to modify the metal–oxygen bond strength in order to optimize both activity and N₂ selectivity.⁴² Note that it has previously been shown that the nature of the reducing agent impacts on catalyst structure and activity.^{43,44} Thus, in this study, the electronic structure of two bimetallic catalysts, Co_BCu/Al₂O₃ and Co_HCu/Al₂O₃, prepared using NaBH₄ and H₂, respectively, are studied with a range of X-ray spectroscopic techniques.

2. Role of Monodispersed Co Atoms: The "monodispersed" Co atoms appear to play a critical role in enhancing the catalytic activity of Co_BCu/Al₂O₃ compared to CuO/Al₂O₃ in Figure 3a. The manuscript should elucidate the specific contributions of cobalt to this improvement. For example, if the Co

can help to disperse CuO or change the reaction pathway?

The role of monodispersed Co atoms is to elongate (weaken) the Cu-O bonds in $\text{Co}_B\text{Cu}/\text{Al}_2\text{O}_3$. Three techniques (including STEM, EXAFS fitting and WT-EXAFS analysis) consistently demonstrate that Cu-O bond length in $\text{Co}_B\text{Cu}/\text{Al}_2\text{O}_3$ ($1.96 \text{ \AA} \pm 0.005$) is longer than in $\text{Co}_H\text{Cu}/\text{Al}_2\text{O}_3$ ($1.93 \text{ \AA} \pm 0.003$). Given the Co-O bond length in $\text{Co}_B\text{Cu}/\text{Al}_2\text{O}_3$ is longer ($2.04 \text{ \AA} \pm 0.0002$) than Cu-O bonds, we propose that Co doping into the CuO lattice elongates the Cu-O bonds. This bond lengthening reduces the Cu-O bond dissociation energy (as evidenced by XPS, HERFD-XANES, and VtC-XES), enhances the redox ability of CuO (supported by operando XAFS and in-situ NAP-NEXAFS), and ultimately improves catalytic activity. The absence of sharp changes in the Co spectra suggests that Cu species are likely the active catalytic sites in the NH_3 -SCO reaction. Much of this has been addressed in our responses to the other reviewers, but we also added the following discussion to the manuscript:

The redox activity of the single Co sites in $\text{Co}_B\text{Cu}/\text{Al}_2\text{O}_3$ and $\text{Co}_H\text{Cu}/\text{Al}_2\text{O}_3$ is similar (Fig. 4f). The single Co sites comprise Co^{2+} at room temperature and are partially oxidized to Co_3O_4 as the temperature increases.

3. The author notes that the selectivity to N_2 for $\text{Co}_B\text{Cu}/\text{Al}_2\text{O}_3$ is greater than 80% at high temperatures, whereas $\text{Co}_H\text{Cu}/\text{Al}_2\text{O}_3$ shows 70% selectivity at 450 °C, along with increased production of NO and N_2O (Fig. 3b, d). Given that the T100 for fresh $\text{Co}_B\text{Cu}/\text{Al}_2\text{O}_3$ is approximately 250 °C and for $\text{Co}_H\text{Cu}/\text{Al}_2\text{O}_3$ is around 350 °C, why is the comparison of N_2 selectivity made at 450 °C? The conclusion of N_2 selectivity between two samples is inaccurate.

Thank you for the suggestion. We compared the activity and selectivity of $\text{Co}_B\text{Cu}/\text{Al}_2\text{O}_3$ and $\text{Co}_H\text{Cu}/\text{Al}_2\text{O}_3$ at the same temperature range for a clear comparison and we have added the following discussion:

Additionally, the selectivity to N_2 for $\text{Co}_B\text{Cu}/\text{Al}_2\text{O}_3$ was $> 80\%$, even at high temperatures, and is lower than $\text{Co}_H\text{Cu}/\text{Al}_2\text{O}_3$ below 350 °C, but is superior to $\text{Co}_H\text{Cu}/\text{Al}_2\text{O}_3$ above 350 °C (Fig. 3b, d).

4. The T100 for NH_3 in $\text{Co}_B\text{Cu}/\text{Al}_2\text{O}_3$ increases from ~ 250 °C to ~ 300 °C, while N_2 selectivity appears to improve from 80-85% to 85-95%. Is this enhancement attributed to the agglomeration of Co species or CuO due to weak metal-support interactions (MSI) with Al_2O_3 prepared by the NaNH_4 reduction method? Does the H_2 -reduced $\text{Co}_H\text{Cu}/\text{Al}_2\text{O}_3$ exhibit similar performance changes?

We have modified the stability test method as suggested, adding 100 mg SiC with the catalysts to avoid hotspot formation. The stability of the catalyst improved, with the T100 for NH_3 using $\text{Co}_B\text{Cu}/\text{Al}_2\text{O}_3$ increasing with enhanced N_2 selectivity. In the operando XAS Co K-edge, there are no Co-Co features

in R-space. Thus, it is possible that weak metal-support interactions cause the changes in performance. We added the following discussion to the manuscript:

Furthermore, $\text{Co}_B\text{Cu}/\text{Al}_2\text{O}_3$ demonstrates good stability under the reaction conditions, showing no significant decrease in catalytic activity or N_2 selectivity even after four consecutive reaction cycles. The reduced activity might be due to weak interactions between the nanoparticles and support material.

5. The author mentions that “Operando DRIFTS confirms different adsorption behavior in $\text{Co}_B\text{Cu}/\text{Al}_2\text{O}_3$ and $\text{Co}_H\text{Cu}/\text{Al}_2\text{O}_3$ (Fig. 6).” However, Figure 6b presents DRIFTS data for $\text{CuO}/\text{Al}_2\text{O}_3$, not for $\text{Co}_H\text{Cu}/\text{Al}_2\text{O}_3$, rendering this description inaccurate.

We sincerely appreciate the reviewer's thorough evaluation. This error has been corrected as suggested. We performed DRIFTS experiments on $\text{Co}_H\text{Cu}/\text{Al}_2\text{O}_3$ and updated Fig. 6. We have added the following discussion to the manuscript:

In-situ DRIFTS confirms different adsorption behaviour in $\text{Co}_B\text{Cu}/\text{Al}_2\text{O}_3$ and $\text{Co}_H\text{Cu}/\text{Al}_2\text{O}_3$ (Fig. 6).⁵⁶ The peak at 1460 cm^{-1} originates from Brønsted acid sites on Al_2O_3 . A peak at 1580 cm^{-1} gradually emerges at temperatures above 250 °C in $\text{Co}_B\text{Cu}/\text{Al}_2\text{O}_3$, which may be tentatively assigned to nitrate species, with this peak only emerging at temperatures above 300 °C in the presence of $\text{Co}_B\text{Cu}/\text{Al}_2\text{O}_3$.

Figure 6 *In-situ* DRIFTS spectra. *In-situ* DRIFTS spectra of $\text{Co}_B\text{Cu}/\text{Al}_2\text{O}_3$ (a) and $\text{Co}_H\text{Cu}/\text{Al}_2\text{O}_3$ (b) as a function of temperature (the catalysts were exposed to a flow of 5000 ppm NH_3 and $5\% \text{ O}_2$ for 20 min at different temperatures).

6. The use of 5000 ppm NO_x without pelleting or dilution sand may lead to hotspot formation in the catalysts during $\text{NH}_3\text{-SCO}$ reaction. Such hotspots can result in elevated NO_x emissions and reduced cycle stability.

Thank you for this excellent suggestion. We have modified the stability test method, adding 100 mg SiC with the catalysts to avoid hotspot formation. The

stability of the catalyst improved, see revised Fig. 3h&i.

Figure 3 Evaluation of Co_BCu/Al₂O₃, Co_HCu/Al₂O₃ and control catalysts in the NH₃-SCO reaction. NH₃ conversion (a) and selectivity to N₂ (b) as a function of temperature (note that CoCu/Al₂O₃ corresponds to the catalyst without reduction); Product distribution at 200 °C (c) and 450 °C (d); e Activation energy for the Co_BCu/Al₂O₃ and Co_HCu/Al₂O₃ catalyzed reaction; Reaction order for NH₃ (f) and O₂ (g); Stability tests of Co_BCu/Al₂O₃ over 4 cycles (h, i) (50 mg Co_BCu/Al₂O₃ mixed with 100 SiC). Reaction conditions: 50 mg catalyst, 5000 ppm NH₃, 5% O₂ balanced in He, gas flow: 100 mL/min, WHSV = 600 mL NH₃·h⁻¹·g⁻¹.

Stability test: 50 mg Co_BCu/Al₂O₃ and 100 mg SiC was mixed by grinding and then tested for four cycles.

Nature Communications manuscript ID: NCOMMS-25-07784-A

Title: Lowering the Cu-O bond energy in CuO nanocatalysts enhances the efficiency of NH₃ oxidation

Note: reviewers' comments appear in **black text**. Our replies appear in **blue text**, and the revised text in the manuscript and supporting information appears in **green**.

Reviewer Comments and point-to-point answers:

Reviewer #1:

The authors revised this manuscript appropriately according to the reviewer's comments. However, there are some issues should be addressed.

We appreciate the reviewer's suggestions, and we have made appropriate modifications. Below are point-to-point responses that address all the reviewer's concerns:

1. Authors thought that "Earth-abundant CoO_x offers a cost-effective alternative. And the overall activity was enhanced by combined with CuO in catalysts, but leads to the formation of NO or N₂O, so lower N₂ selectivity." However, the poor N₂ selectivity can be observed on Co/Al₂O₃ from Fig. 3b, Cu/Al₂O₃ displayed the highest N₂ selectivity among the investigated catalysts. Therefore, the N₂ selectivity of Cu-Co catalysts were improved due to the introduction of Cu into Co/Al₂O₃.

We have reworded the sentence as below.

Earth-abundant CoO_x offers a cost-effective alternative, with the introduction of Cu into CoO_x improving the N₂ selectivity of CoO_x based catalysts.^{35,36}

2. Authors suggested that "weaker metal–oxygen bonds tend to exhibit higher rates of reaction, but result in the formation of NO_x (NO and N₂O). Compared to Co_HCu/Al₂O₃, the superior activity and lower selectivity to N₂ observed for Co_BCu/Al₂O₃ may be attributed to the electronic structure of Cu²⁺ produced by the local coordination environment (i.e. weaker Cu–O bonding)." But there was no data on the bond energy of metal-oxygen bonds in the revised manuscript, which should be supplemented.

Thank you for the suggestions. While we cannot directly determine the energy of Cu-O bond with and without Co doping, the combination of XAS/XES data present the trends of bond energy changes. In principle, XAS (HERDF-XANES in our manuscript) measures the valence unoccupied states, which corresponds to the $\sigma_{\text{Cu}4\text{p}-\text{O}2\text{p}}$ antibonding states and Cu_{3d}-O_{2p} nonbonding states. Vtc-XES probes the valence occupied states, i.e. the $\sigma_{\text{Cu}4\text{p}}$ -

o_{2p} bonding states. We plot the combination of XAS/XES spectra in Fig. S7 with the energy gap between XAS and XES features representative of the trends in bond energies.

We also discussed why this gap is not the real bond energy, see below. The accurate Cu–O bond energy cannot be derived from our data. In XANES, the intrinsic lifetime broadening of the $1s$ hole ($\sim 1\text{--}2$ eV) significantly smears out fine structures. HERFD-XANES reduces this broadening by selecting a narrow emission channel (typically the Cu $K_{\alpha 1}$ line), leaving only the $2p_{3/2}$ core-hole width (~ 0.4 eV). The creation of the core hole leads to an increased local Coulomb potential, which alters the final-state electronic structure through screening by valence and ligand electrons. This interaction gives rise to chemical shifts and modifies the relative intensity of near-edge features, particularly the pre-edge and white-line region. Thus, we cannot use them to calculate Cu–O bond energy. Even so, XANES and VtC results representing the $1s \rightarrow 4p_z$ antibonding transition and $4p \rightarrow 1s$ bonding transition of Cu–O, respectively. And such transitions are related to Cu–O bond energy. Thus, we are able to conclude that the $\text{Co}_B\text{Cu}/\text{Al}_2\text{O}_3$ has weaker Cu–O bonds.

Catalysts with weaker metal–oxygen bonds tend to exhibit higher rates of reaction, but result in the formation of NO_x (NO and N_2O). Compared to $\text{Co}_H\text{Cu}/\text{Al}_2\text{O}_3$, $\text{Co}_B\text{Cu}/\text{Al}_2\text{O}_3$ has a lower $1s \rightarrow 4p_z$ antibonding transition and a higher $4p \rightarrow 1s$ bonding transition of Cu–O bonds, indicative of weaker Cu–O bonds.^{54–56} Thus, the superior activity and lower selectivity to N_2 observed for $\text{Co}_B\text{Cu}/\text{Al}_2\text{O}_3$ may be attributed to the electronic structure of Cu^{2+} produced by the local coordination environment.

Fig. S7. **b** VtC and XANES of $\text{CuO}/\text{Al}_2\text{O}_3$, $\text{Co}_B\text{Cu}/\text{Al}_2\text{O}_3$ and $\text{Co}_H\text{Cu}/\text{Al}_2\text{O}_3$.

Reviewer #2

The authors have conducted a thorough revision and answered all my questions and incorporated all the suggestions. I do not have any further comments and recommend the article to publish.

Thank you.

The authors should conduct a proper spelling check of the manuscript. For example, caption of Figure 1, instead of HADDF it should be HAADF.

We have carefully checked the manuscript and hopefully corrected all abbreviations and spelling mistakes.

Reviewer #3

The authors have well addressed my earlier comments. Acceptance is therefore recommended.

Thank you.